# Daily rhythm in cortical chloride homeostasis underpins functional changes in visual cortex excitability

Enrico Pracucci [1,8], Robert T. Graham [2,8], Laura Alberio [2,8], Gabriele Nardi [1,8], Olga Cozzolino[1], Vinoshene Pillai [1], Giacomo Pasquini [1], Luciano Saieva[2], Darren Walsh [2], Silvia Landi [3,7], Jinwei Zhang [4,5], Andrew J. Trevelyan [2,9] ✉ & Gian-Michele Ratto [1,3,6,9] ✉

Cortical activity patterns are strongly modulated by fast synaptic inhibition mediated through ionotropic, chloride-conducting receptors. Consequently, chloride homeostasis is ideally placed to regulate activity. We therefore investigated the stability of baseline $[Cl^-]_i$ in adult mouse neocortex, using in vivo two-photon imaging. We found a two-fold increase in baseline $[Cl^-]_i$ in layer 2/3 pyramidal neurons, from day to night, with marked effects upon both physiological cortical processing and seizure susceptibility. Importantly, the night-time activity can be converted to the day-time pattern by local inhibition of NKCC1, while inhibition of KCC2 converts day-time $[Cl^-]_i$ towards night-time levels. Changes in the surface expression and phosphorylation of the cation-chloride cotransporters, NKCC1 and KCC2, matched these pharmacological effects. When we extended the dark period by 4 h, mice remained active, but $[Cl^-]_i$ was modulated as for animals in normal light cycles. Our data thus demonstrate a daily $[Cl^-]_i$ modulation with complex effects on cortical excitability.

Brain function varies markedly through the day, as evident from the strong daily cyclical patterns seen in many functional neurological and psychiatric conditions[1–6]. Basic animal research has further documented variation in the level of neuronal firing rates[7–9], in the number, structure, strength and plasticity of synapses[10–15], in metabolic state[16], gene expression and protein phosphorylation[17,18], and in the extracellular environment of neurons[19] at different times of day. Despite the extensive evidence linking the circadian clock and cortical function[20], we know little about the actual mechanistic basis of this relationship.

A key determinant of neuronal activity is the distribution of the major ionic species between the intra and extracellular spaces. A large body of work has examined rapid, activity-dependent redistribution of ions (sometimes termed "ionic plasticity"[21]), but little in vivo data exists regarding baseline levels, and whether these are modulated during the day. Of significant interest, therefore, is the recent demonstration in multiple different non-neuronal cell classes of redistribution of $Na^+$, $K^+$ and $Cl^-$ between the intra and extracellular space, to offset fluctuations in macromolecular crowding. This crowding-associated compensatory

[1]National Enterprise for nanoScience and nanoTechnology (NEST), Istituto Nanoscienze, Consiglio Nazionale delle Ricerche (CNR) and Scuola Normale Superiore Pisa, 56127 Pisa, Italy. [2]Newcastle University Biosciences Institute, Medical School, Framlington Place, Newcastle upon Tyne NE2 4HH, UK. [3]Institute of Neuroscience CNR, Pisa, Italy. [4]Institute of Biomedical and Clinical Sciences, Medical School, College of Medicine and Institute of Health, University of Exeter, Hatherly Laboratories, Exeter EX4 4PS, UK. [5]State Key Laboratory of Chemical Biology. Research Center of Chemical Kinomics, Shangai. Institute of Organic Chemistry, Chinese Academy of Sciences, Shanghai 200032, China. [6]Padova Neuroscience Center, Padova, Italy. [7]Present address: National Enterprise for nanoScience and nanoTechnology (NEST), Istituto Nanoscienze, Consiglio Nazionale delle Ricerche (CNR) and Scuola Normale Superiore Pisa, 56127 Pisa, Italy. [8]These authors contributed equally: Enrico Pracucci, Robert T. Graham, Laura Alberio, Gabriele Nardi. [9]These authors jointly supervised this work: Andrew J. Trevelyan, Gian-Michele Ratto. ✉e-mail: andrew.trevelyan@newcastle.ac.uk; gianmichele.ratto@sns.it

transport of ions involves slow baseline changes of ionic concentration to accommodate circadian regulation of proteome renewal whilst maintaining osmotic balance and cell volume[22]. It is unknown if this process also occurs in neurons, where it could provide a slow periodic change in cellular ion content and excitability, on top of which lies the more transient ionic variation due to neuronal activity. The existence of this dual modality in ionic homeostasis has never been examined in neuronal cells, where osmotic regulation and ionic regulation may be prioritized differently.

In this study, we focused upon intracellular Cl⁻ concentration ($[Cl^-]_i$). The importance of $[Cl^-]_i$ derives from the fact that fast synaptic inhibition in mammalian brains is subserved primarily by GABA acting upon ionotropic, chloride-conducting GABA$_A$ receptors (GABA$_A$-Rs)[23–25]. $[Cl^-]_i$ is tightly regulated by the equilibrium of fluxes through ion channels and two key co-transporters: KCC2, that normally extrudes Cl⁻ and K⁺, and NKCC1, that moves in two Cl⁻ ions together with one K⁺ and one Na⁺ ion. $[Cl^-]_i$ determines the GABAergic reversal potential ($E_{GABA}$) that typically lies close to resting membrane potential, but relatively small rises in $[Cl^-]_i$ can push $E_{GABA}$ close to, or even above, action potential threshold. Such excitatory effects of GABA have been shown during epileptic discharges[26] and in early cortical development[27–29]. Since GABAergic function dictates so many different neuronal network functions and states[30–38], $[Cl^-]_i$ is ideally placed to be a regulator of neuronal activity. In the adult brain, $[Cl^-]_i$ in neurons has generally been considered to be stable, with $E_{GABA}$ substantially below action potential threshold. Chloride levels may rise acutely, following short bursts of GABAergic synaptic activity[39], and can also be loaded into neurons artificially[33,40], but in both cases, the baseline chloride levels are rapidly re-established through the action of KCC2[27,33]. In the adult brain, persistently raised $[Cl^-]_i$ in cortical neurons was only thought to occur in association with pathophysiology[26,28,33,41–46]. The difficulty of making the requisite measurements in populations of neurons in vivo has meant that this assumption has never been examined explicitly. The development of LSSmClopHensor for use in vivo[28,47,48] now makes this achievable. We performed LSSmClopHensor imaging in young adult mice and report a hitherto unrecognized and unexpectedly large fluctuation in baseline $[Cl^-]_i$, in neocortical pyramidal cells over the day / night cycle. These data suggest that GABAergic function varies during the day, and we confirmed this prediction using assays of network excitability. Indeed, we found greatly increased excitability at night, when $[Cl^-]_i$ is high. We further relate these findings to changes in both membrane expression and functionality of the cation-chloride co-transporters. Accordingly, the night-time results were consistently shifted towards those seen in the day, by acute inhibition of NKCC1, while the inhibition of KCC2 during the day raised excitability. Interestingly, mice remained active by extending the dark period by 4 h, but the cycle of $[Cl^-]_i$ modulation was unchanged.

## Results

### In vivo 2-photon imaging of intracellular $[Cl^-]$ and pH in pyramidal neurons

We introduced the pH and chloride biosensor LSSmClopHensor[28,47] into neocortical pyramidal cells, in two different ways: either by in utero electroporation at embryonic day 15.5, using a plasmid encoding LSSmClopHensor under the CAG promoter, or utilizing a viral vector to deliver a floxed construct of LSSmClopHensor in mice expressing Cre recombinase under the *Emx1* promoter (see Methods for details). Both the *in utero* electroporation[49] and the viral vector (Supplementary Fig. S1) reliably induced widespread labelling of the main population of supragranular (layer 2/3) excitatory neurons in the occipital cortex of young mice (1–4 months old), which may be visualized directly through a cranial window using 2-photon microscopy.

Mice were kept on a 12-h light / 12-h dark cycle, switching at 7:00 and 19:00 h. The start of the light period (7:00 h) is termed *zeitgeber*

time 0 (ZT0; 19:00 h is ZT12), when the mice bed down and sleep[50]. At least 3 days prior to all experimental procedures, we moved the mice into a quiet room, where they were checked either solely by video, or by direct visualization no more than once a day, and not handled, to minimize disturbance to their daily cycle. In all cases, mice showed a peak of activity in the hour immediately after the lights were turned off, and were typically active throughout the dark period, albeit with short periods of quiescence. In contrast, mice rarely moved from their nest during the light periods (Fig. 1, and Supplementary Fig. S2).

We performed in vivo LSSmClopHensor imaging of layer 2/3 pyramidal neurons, under anaesthesia, at 4 different times of day: ZT2 (9:00 h), ZT5 (12:00 h), ZT12 (19:00 h) and ZT17 (midnight). Surgeries were started up to an hour before these times, and imaging was performed always within 2 h of the start of the surgery. The procedure involved performing a craniotomy over the labelled hemisphere, which was covered with a glass coverslip and cemented in place, prior to transferring the animal onto a 2-photon microscope imaging platform. We collected data from various fields of view, between 90 and 300 μm deep to the pial surface, which corresponds approximately to layers 2–3. In total, we imaged 2851 neurons, from 35 different mice (Fig. 1, Supplementary Figs. S3–4). There was negligible fluctuation in pH across the different imaging times (Supplementary Fig. S3). In contrast, we found a marked variation in $[Cl^-]$ inside pyramidal cells, over the day/night cycle, with the lowest values occurring around ZT5 (median $[Cl^-]_i = 8.9$ mM), in the middle of the murine resting period, and the highest values at ZT17 (median $[Cl^-]_i = 18.4$ mM; ZT5 vs ZT17, $p = 0.002$ Mann–Whitney test), when the mice were very active (Fig. 1a–c; Supplementary Fig. S4). There was no systematic relationship between the median values of pH and $[Cl^-]_i$, across the population of imaged mice (Supplementary Fig. S5). At ZT2 and ZT12, pyramidal $[Cl^-]_i$ showed intermediate values, but also, interestingly, the largest inter-animal variability, possibly arising from phase differences of the daily cycle between the various mice. The dispersion of $[Cl^-]_i$ values, within each mouse, was minimal during the day but larger during the night (Fig. 1c). There were no differences between male and female animals, nor between the Pisa and Newcastle animals (different strains, method of introducing LSSmClopHensor and anaesthesia), indicating that the daily modulation of pyramidal $[Cl^-]_i$ is a robust observation. Notably, the day-to-night change in pyramidal $[Cl^-]_i$ is the exact opposite cycle to that seen in neurons in the suprachiasmatic nucleus (SCN), the one site in the brain where significant daily modulation of neuronal $[Cl^-]_i$ has been demonstrated previously; there, the $[Cl^-]_i$ is high during the day, and low at night[51].

Next, we asked whether the marked reduction of $[Cl^-]_i$ that occurs between ZT17 and ZT5 was triggered by the light onset at ZT0. To answer that, mice were placed under videorecording for 3 days maintaining the light cycle from 7:00 (light on) until 19:00 (light off). On the day of imaging, the light was left off at ZT0 (7:00) and therefore mice underwent over 16 h of darkness, before being imaged at ZT28-29. All the mice remained active during this extended period of darkness (Fig. 1e). In 4 out of 5 animals, the periods of immobility were far lower (Fig. 1f) indicating that sleep was markedly reduced during prolonged darkness, even if this cannot be considered to be complete sleep deprivation. Despite these changes in behaviour, notably, the $[Cl^-]_i$ fell in line with the modulation seen in animals under normal light cycles (Fig. 1b, d). This suggests that the exposure to light at ZT0, and the subsequent strong reduction in the activity of the mice, are not causally related to the observed $[Cl^-]_i$ cycle.

Elevated $[Cl^-]_i$ within pyramidal neurons, in early postnatal life, can be countered by the focal application of bumetanide, an inhibitor of the Na⁺-K⁺-Cl⁻-cotransporter NKCC1[28]. We therefore examined whether the nocturnal elevation of $[Cl^-]_i$ in our mice could similarly be altered. Following a single treatment, the pharmacological effect of bumetanide stabilized in about 40 min. (Supplementary Fig. S6). Imaging was performed starting from 40 min from the beginning of the treatment and

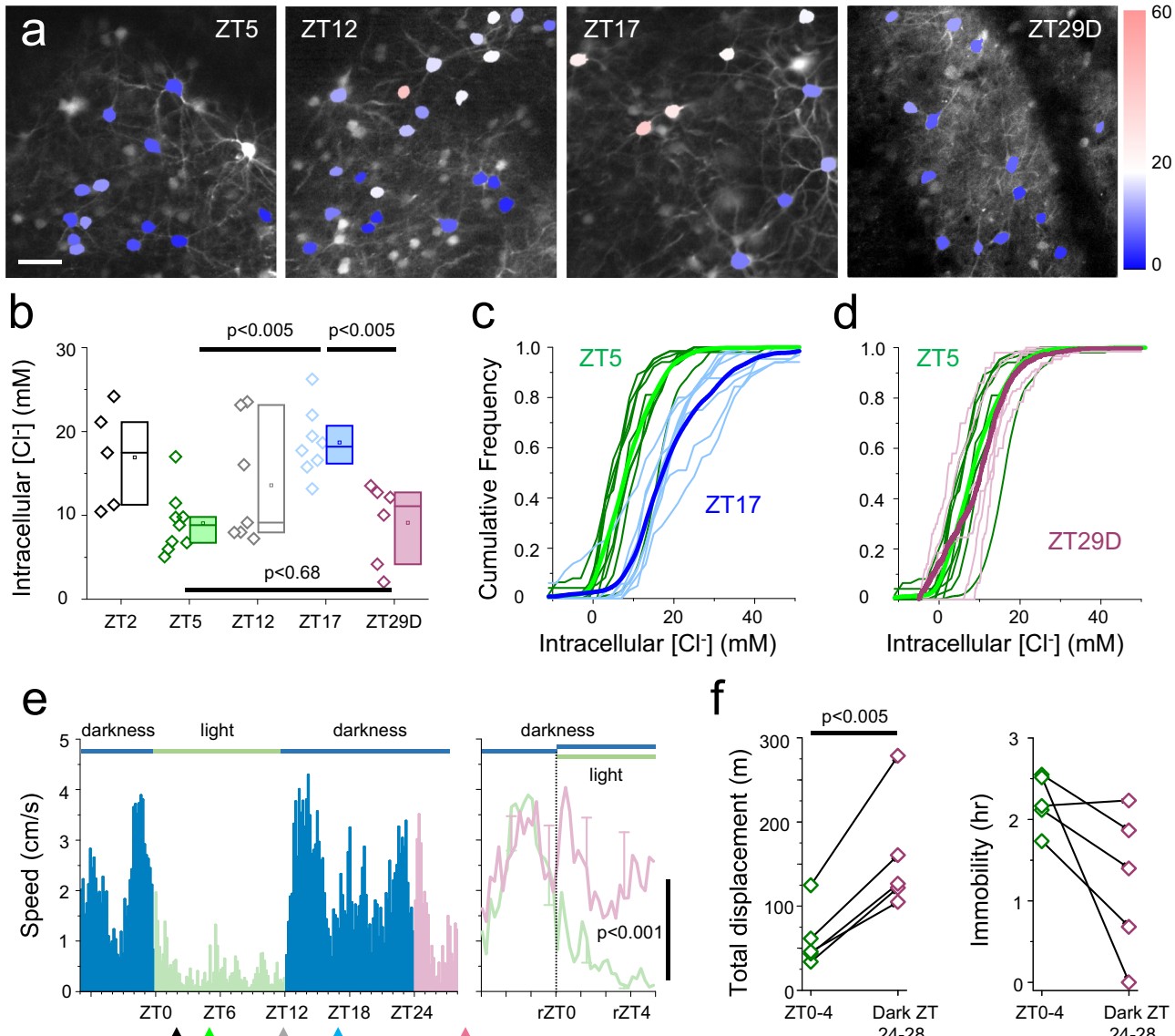

**Fig. 1 | Daily change of intracellular Chloride in cortical pyramidal neurons.**
**a** Four representative fields of view, showing pyramidal cells expressing LSSmClopHensor, in layers 2/3 of visual cortex, in mice aged >1 month. Mice were imaged, under terminal anaesthesia, at the indicated times. The fourth field, labelled ZT29D, has been imaged after leaving the mice in darkness from ZT12 until anaesthesia 16 h later. Calibration bar 40 μm. An average of 4 fields have been acquired for each mouse. **b** Quantification of $[Cl^-]_i$ at 5 time points: ZT2 (9:00, 5 mice and 159 neurons), ZT5 (12:00, 9 mice and 663 neurons), ZT12 (19:00, 7 mice and 390 neurons), ZT17 (24:00, 8 mice and 1051 neurons) and ZT29 (6 mice, 588 neurons). The points are the median values for individual animals (see also Supplementary Fig. S4), and the box plots show the interquartile range, median (central line) and mean (small symbol) for each experimental group. There is no difference between ZT5 and ZT29D, but both are significantly different from ZT17 (Mann–Whitney, two-sided. ZT5 vs ZT17: $p = 1.8*10^{-3}$; ZT17 vs ZT29D: $p = 3.7*10^{-3}$; ZT5 vs ZT29D: $p = 0.68$). **c** Cumulative distribution for each mouse imaged at ZT5

(green traces) and at ZT17 (blue traces). The thick lines show the mean distributions of the two groups. **d** Cumulative distribution for each mouse imaged at ZT5 (green traces) and at ZT29 after continue darkness starting from ZT12 (magenta traces). **e** Actogram obtained from the continuous videorecording of 5 out of the 6 mice shown in (**b**) at the ZT29D point (only 5 mice are included because the video recording malfunctioned for one animal). The mice kept in prolonged darkness were more active compared to the equivalent period when they were in the light (see magnification in the right panel. paired t-test, two-sided, $p = 1.8*10^{-4}$. The plot shows the mean ± variance. Bin duration: 10 min. **f** The cumulative distance travelled by each mouse increased in prolonged darkness (, paired t-test, two-sided, $p = 3.5*10^{-3}$, $n = 5/5$), and the time spent in complete immobility decreased in 4 out of 5 mice. There is no significant correlation between the distance travelled and individual median $[Cl^-]_i$ ($p = 0.47$, ANOVA, $n = 5$). Source data are provided as separate sheets in the Source Data file.

lasted for about 30 min. Local cortical application of 55 μM bumetanide had almost no effect on $[Cl^-]_i$ when applied during the day (Fig. 2a, b, median change in $[Cl^-]_i$ before and after bumetanide, $\Delta[Cl^-]_i = 0.29$ mM, $p = 0.047$, $n = 5$; Wilcoxon signed rank test), but it markedly reduced $[Cl^-]_i$ ZT17 (median $\Delta[Cl^-]_i = -8.7$ mM, $p < 0.001$, Wilcoxon signed rank, Fig. 2b). The difference between the two time points is highly significant ($p < 0.001$, Mann–Whitney, Fig. 2b, Supplementary Fig. S6). The effect of bumetanide at ZT17 is summarized in Fig. 2c. Bumetanide had no

effect on intracellular pH either at ZT5 or ZT17 (ZT5 pre-bumetanide (196 cells, 4 mice), pH = 7.08, post-bumetanide (162 cells, 4 mice), pH = 7.22, $p = 0.11$; ZT17 pre-bumetanide (149 cells, 5 mice), pH = 7.23, post-bumetanide (162 cells, 4 mice), pH = 7.22, $p = 0.78$, Wilcoxon signed rank test). There was also no statistical difference in pH between ZT5 and ZT17 measurements ($p = 0.07$).

Conversely, we should expect that the inhibition of KCC2 during the day would lead to an increase of $[Cl^-]_i$. To test this, we made a focal

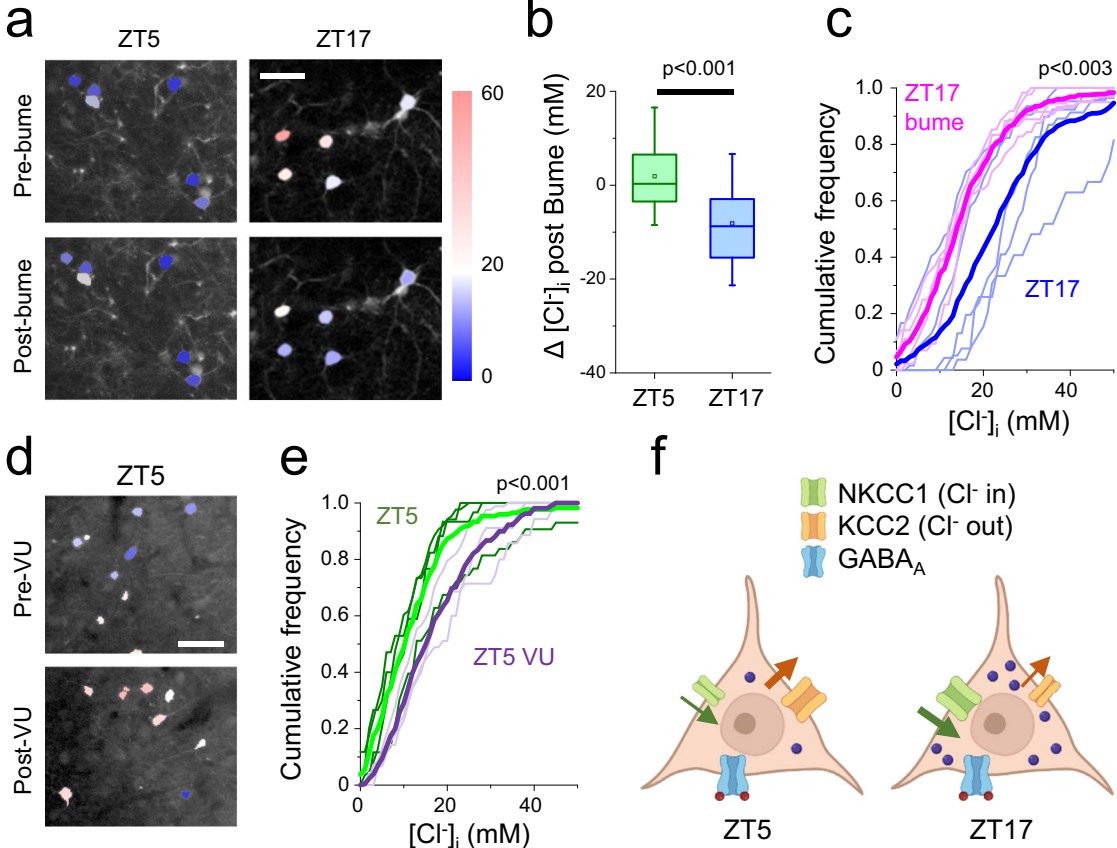

**Fig. 2 | Differential role of Cl⁻ co-transporter in baseline Cl⁻ regulation.**
**a** Representative fields imaged at ZT5 and ZT17 before, and 40 min after, topical application of the NKCC1 inhibitor bumetanide (bume, 55 μM): $[Cl^-]_i$ decreased in all neurons in the field. Calibration bar 40 μm. The colour bar indicates $[Cl^-]_i$ in mM. **b** Change of $[Cl^-]_i$ measured in the same neurons before and after bumetanide treatment. Data from 4 mice (188 neurons) imaged at ZT5, and 5 mice (218 neurons) imaged at ZT17. Box plots show the interquartile range, median (central line) and mean (small symbol). Whiskers extend down to the 5th centile and up to the 95th centile. ($p = 2.2*10^{-16}$, Mann–Whitney, two-sided). **c** Cumulative frequency distributions of the effects of bumetanide treatment on $[Cl^-]_i$ at ZT17. The thin lines are the distributions from individual mice and the thick lines are the average distributions (Mann–Whitney, two-sided, $p = 7.5*10^{-6}$). **d** Two representative fields

imaged at ZT5, before and after topical application of the KCC2 inhibitor VU0463271 (VU, 10 μM). Calibration bar 40 μm. **e** Effects of the VU0463271 treatment on all imaged neurons at ZT5 (control 4 mice, 176 neurons; VU0463271 3 mice, 163 neurons, Mann–Whitney, two-sided, $p = 4.7*10^{-5}$). The thin lines are the distributions from individual mice and the thick lines are the average distributions. **f** Schematic representation of the suggested mechanism for the daily change in $[Cl^-]_i$. During the day (rest phase) the baseline $[Cl^-]_i$ is mainly determined by the extrusion mediated by KCC2 and the inhibition of NKCC1 does not produce a discernible effect. In contrast, at night, the opposite picture is true, as the higher activity of NKCC1 raises the $[Cl^-]_i$ baseline level. Source data are provided in the Source Data file.

application of 10 μM of the KCC2 inhibitor VU0463271 at ZT5, and found that this did indeed cause $[Cl^-]_i$ to rise (Fig. 2d, e). In summary, our data suggest that during the day, low $[Cl^-]_i$ is maintained by high activity of the exporter KCC2; but, at night, NKCC1 becomes dominant, thereby raising the baseline $[Cl^-]_i$ (Fig. 2f). Together, these pharmacological assays indicate that the daily variation in $[Cl^-]_i$ reflects a shift in the relative activity of the two cation-chloride cotransporters found in these cells.

### Daily variation in chloride-cation co-transporter expression and phosphorylation

The imaging and pharmacological data suggest that the daily cycle of $[Cl^-]_i$ may be linked to changes in the relative contribution of NKCC1 and KCC2 to chloride regulation[52]. These are the only chloride-cation cotransporters expressed in neurons and their function is known to be bidirectionally regulated by protein kinases. KCC2 activity is enhanced by phosphorylation of Ser940[29,53] and inhibited by phosphorylation of Thr1007[54], causing internalization of the cotransporter from the membrane, which may be assayed by measuring the proportion of the total KCC2 that is retained in the membrane. In contrast, NKCC1 activity is upregulated by phosphorylation at Thr203/Thr207/Thr212[55].

Therefore, we examined the expression levels and phosphorylation state of KCC2 and NKCC1 during the daily cycle, using antibodies that have been extensively characterized in our previous work[56–58] (Fig. 3, Supplementary Fig. S7).

We first utilized whole-cell biotinylation followed by immunoblotting of neocortical tissue that had been flash-frozen following an acute dissection from freshly sacrificed mice at either ZT5 or ZT17 (in our sampling, the times showing the maximal $[Cl^-]_i$ difference). The biochemical analysis was performed blind to the experimental grouping. Analysis of immunoblotting showed no difference in the total KCC2 protein levels (Students' $t$-test, $n = 4$ mice; $p = 0.82$, Fig. 3b), but there was a significant reduction in the surface KCC2 protein levels (by 50.2%, Students' $t$-test, $n = 4$ mice; $p = 0.032$, Fig. 3b) between ZT5 and ZT17, indicating that there was a marked redistribution of the protein away from the plasma membrane at night. We further examined whether there were alterations of phosphorylation state of either KCC2 or NKCC1 between day and night, indicative of changes in functional activity. We found that KCC2 showed no alteration in phosphorylation at Ser940 ($n = 4$ mice; $p = 0.98$, Fig. 3b), but a significant increased level, at ZT17, of KCC2 Thr1007 phosphorylation (by 53%, $n = 4$ mice; $p = 0.036$, Fig. 3b), and

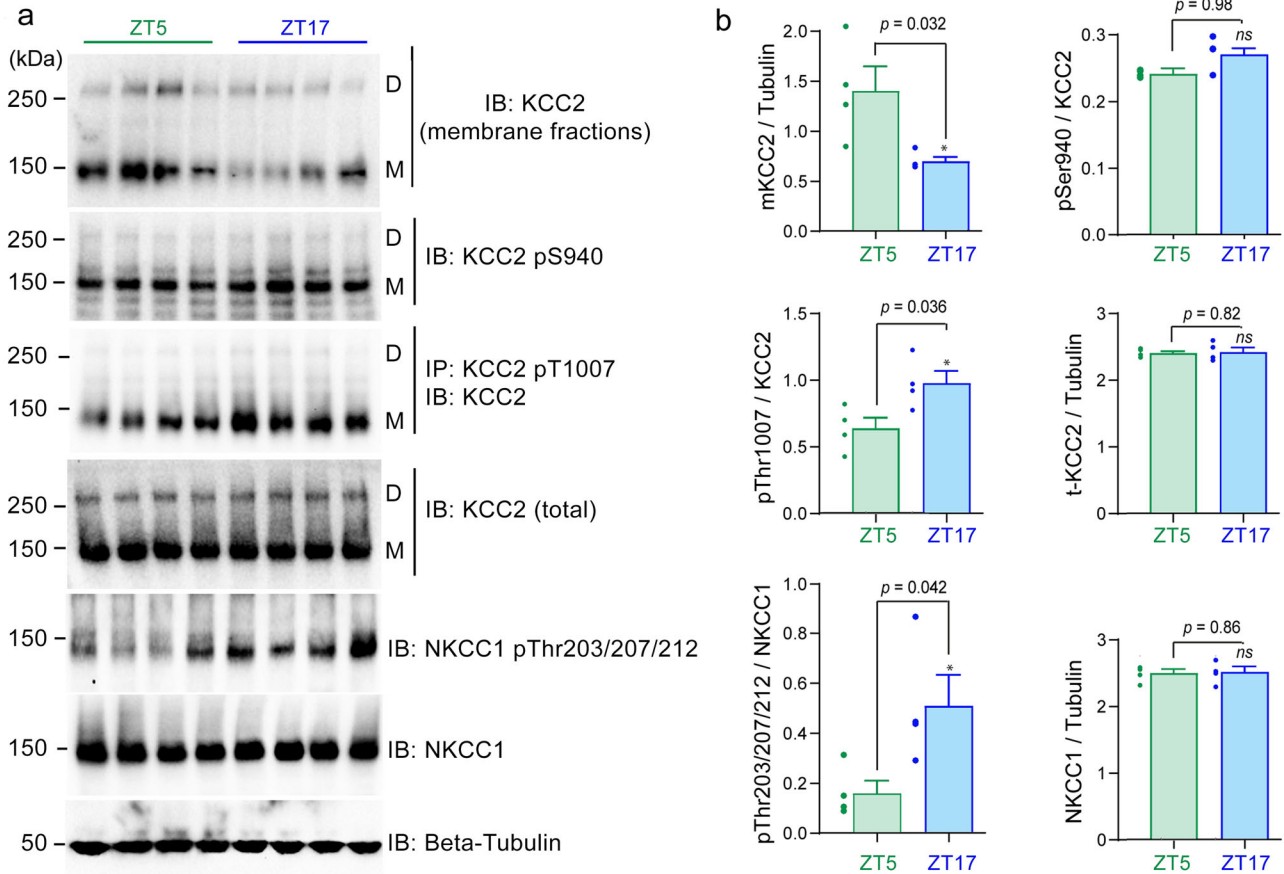

**Fig. 3 | Changes in KCC2 membrane stability and the phosphorylation of KCC2 and NKCC1 in cerebral cortex during the daily cycle. a** Top panel shows biotinylated surface fraction and total protein expression of KCC2 in membrane fractions prepared from flash-frozen neocortex at different times of day. For following panels: harvested cerebral cortex lysates were subjected to biotinylated surface fraction and total protein expression of KCC2, or immunoprecipitation (IP) with the indicated KCC2 Thr[1007] phospho-antibody and the immunoprecipitants were then immunoblotted (IB) with the indicated specific KCC2 antibody. Cerebral cortex lysates were also subjected to IB analysis with the indicated total and phospho-specific antibodies. Both KCC2 dimers (D) and KCC2 monomers (M) are indicated. Molecular masses are indicated in kDa on the left-hand side of the Western blots. β-tubulin immunoblots were used as a loading control in each gel, and a representative example is shown at the bottom. Western blots are from the same quantified samples. **b** The right panel shows quantification of the results of the Western blots, as assessed by Students' $t$-test, two-sided ($n = 4$, error bars represent the mean ± SEM., four independent experiments). The quantification (ratio calculation) is illustrated in Supplementary Fig. 7.Source data and uncropped scans are provided as separate sheets in the Source Data file.

NKCC1 Thr203/Thr207/Thr212 phosphorylation (by 220%, $n = 4$ mice; $p = 0.042$, Fig. 3b) was detected. These data therefore provide two different mechanisms which would be expected to lead to increased [Cl⁻]ᵢ at night, namely (1) a decrease in the cell surface expression of KCC2 following the phosphorylation of KCC2 at the Thr1007 site, (2) phosphorylation of NKCC1 at sites which upregulate its activity. While these biochemical assays are not cell-specific, the day-to-night differences are consistent with the observed daily modulation in [Cl⁻]ᵢ, in pyramidal cells and with the pharmacological evidence (Fig. 2).

### Effects of the daily cycle of [Cl⁻]ᵢ on visual processing

We next examined what consequences these changes in [Cl⁻]ᵢ might have for cortical processing. We recorded, in primary visual cortex of awake mice, the visual responses to the presentation of a large drifting grating, which has previously been reported to elicit a robust gamma oscillation mediated through feedback inhibition of somatostatin interneurons[59]. Given the well-defined role of fast GABAergic inhibition onto pyramidal neurons in shaping this activity, we predicted that the oscillations would be more tightly defined during the day, when pyramidal [Cl⁻]ᵢ is low, relative to visually evoked oscillations recorded at night, when pyramidal [Cl⁻]ᵢ is high.

To test this hypothesis, we prepared mice with 200 μm chronic tungsten electrodes implanted in their primary visual cortex (V1), and reference screws placed on the cerebellum surface. After recovery from surgery, mice were trained for three days to freely run on a ball while being head-restrained. Training sessions were performed twice a day, early in the morning (ZT2) and late in the afternoon (ZT10-12), to minimize disruption to the daytime resting period of the mice. Each training session lasted about 10–15 min, and the mice were given food rewards at the end. Mice were recorded on alternate days, at ZT5 and ZT17, allowing for at least 36 h of rest between different sessions. At ZT5, presentation of the drifting grating induced robust oscillations in the 20–30 Hz range (Fig. 4a), like that reported in the original description of this phenomenon[59]. When the same mice were recorded at ZT17, oscillations were markedly less prominent (paired Wilcoxon signed rank test; $p \ll 0.001$). We concluded that visually induced gamma oscillations do indeed vary by the time of the day (Fig. 4c, see Supplementary Fig. S8 for Methodology).

These experiments are consistent with our finding of altered [Cl⁻]ᵢ between ZT5 and ZT17, but are correlative, rather than demonstrating a clear causal link. To test the causal link explicitly, we examined the effect of local application of 55 μM bumetanide that we have demonstrated to be sufficient to reduce [Cl⁻]ᵢ at ZT17, but which had no effect

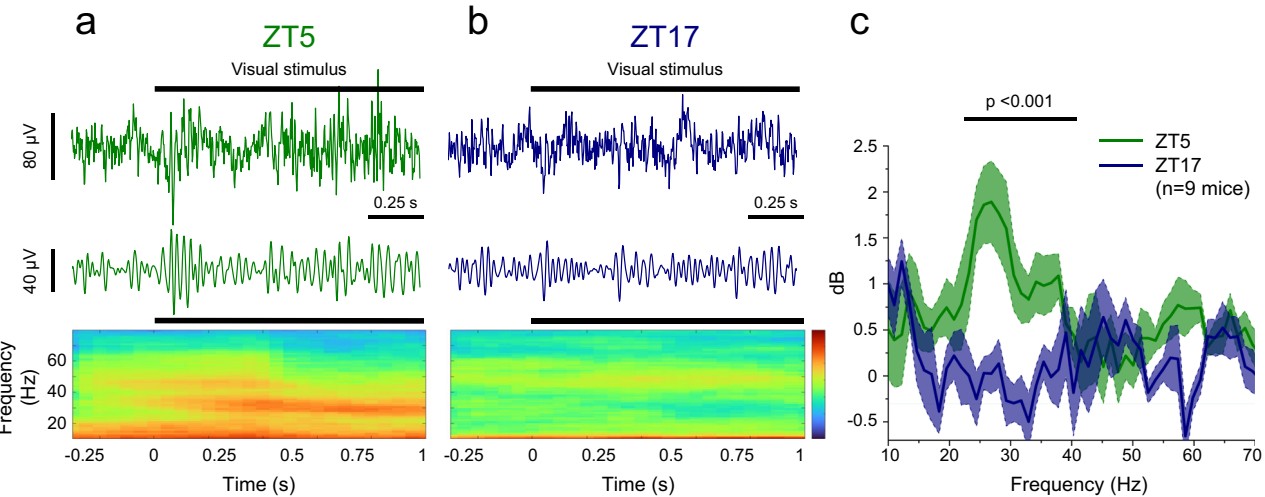

**Fig. 4 | Synchronization of the local field potential in the gamma band depends on the time of the day.** Mice are implanted with a tungsten chronic electrode and are trained for a few days to remain head-fixed on a freely rotating ball. After training, recordings are performed at midday (ZT5, **a**) and midnight (ZT17, **b**). The upper traces show the full band width before and during the grid presentation. The mid trace shows the signal filtered in the 20–60 Hz band. The grid appears at time 0 as indicated under the spectrogram. Colormap limits: [−5,8] dB. **c** Change in the spectral power of the LFP during the period of visual stimulation for all recorded mice (see Methods and also Supplementary Fig. S8 for methodological details). Recordings performed at midday show a narrow band component that is completely missing from the recordings obtained at night (frequency range 20–40 Hz, $p = 3.2*10^{-4}$; Wilcoxon Signed Ranks Test, two-sided). Data pooled from 9 mice recorded at both times of the day and the band shows the SEM. Source data are provided in the Source Data file.

at ZT5 (Fig. 2c). We hypothesized that bumetanide would convert the ZT17 pattern of visually evoked oscillations towards the ZT5 pattern, while having minimal effect upon ZT5 responses. Since these experiments required direct access to the cortex through an open craniotomy, we performed these under urethane anaesthesia. As with our recordings in awake mice, anaesthetised mice also showed prominent visually induced gamma oscillations[59], and consistent with the previous experiments, the peak gamma power was more marked at ZT5 than ZT17 (Fig. 5, median peak prominence at ZT5 = 6.9 dB; at ZT17 = 5.3 dB; Mann–Whitney test: $p = 0.0021$). Notably, bumetanide did not alter the evoked oscillations recorded at ZT5 (Fig. 5a, c–e. Median peak prominence = 6.7 dB. Median increase after bumetanide = 0.18 dB; $p = 0.69$, Wilcoxon signed rank test) but caused a large increase at ZT17 (Fig. 5b, c–e, Supplementary Fig. S9; median peak prominence = 7.2 dB. Median increase after bumetanide = 1.3 dB; $p = 0.016$, Wilcoxon signed rank test). These data are thus consistent with the imaging and biochemical data, indicating that physiological rhythms are modulated according to daily fluctuations of pyramidal $[Cl^-]_i$.

### Effects of the daily cycle of $[Cl^-]_i$ on network stability and epileptic threshold

Raised chloride levels have previously been associated with epileptic pathophysiology, and indeed, have frequently been cited as a primary cause of seizure activity[26,28,33,41–46]. We were interested therefore to examine how the physiological variation in $[Cl^-]_i$ between night and day affected epileptic susceptibility. To do this, we examined the response to an acute, focal 4-aminopyridine (4-AP) injection (500 nl, 15 mM in artificial cerebro-spinal fluid) directly into the neocortex in urethane-anaesthetized mice. We recorded the evolving neocortical field potential (LFP) in experiments performed at either ZT5 or ZT17, (Fig. 6a; see also Supplementary Fig. S10 for methodological details). Mice were recorded for at least 10 min, to ensure a stable electrophysiological baseline, with slow wave oscillations including transient bursts of activity, characteristic of the UP states of non-REM sleep (Fig. 6b). At ZT17, intracortical 4-AP injection induced a rapid build-up of pathological discharges, culminating in a full electrographic seizure in all 8 mice (100% of experiments), with a latency of between 7–29 min after the injection (mean = 17.9 ± 9.5 min SD, Fig. 6d, e; see

Supplementary Fig. S10 for methods). In contrast, at ZT5, 4-AP induced electrographic seizures in only 3 out of 11 mice (Fig. 6d, e; Supplementary Fig. 11 for details of the seizures; recordings were terminated 1 h after 4-AP injection). For the purposes of data collation, if no seizures had occurred by 60 min when recordings were stopped, the latency was designated to be 60 min (8/11 recordings at ZT5 showed no seizures). In this way, we estimated that the ZT5 experimental group had a minimum average latency of 52 min, compared to the ZT17 latency of 17.9 min, which represents a minimum 3-fold increase in latency in the ZT5 versus the ZT17 group (Mann–Whitney test, $p < 0.005$).

To investigate the causal relationship more directly, in a different set of ZT17 experiments, we pretreated the mice with bumetanide, applied directly to the cortical surface for 40 min before the intracortical 4-AP injection. In 5 out of 6 of these bumetanide recordings, no seizures occurred within 1 h, while one mouse had a single, modest seizure after 52 min (Fig. 6d and Supplementary Fig. S11; Mann–Whitney test, night bumetanide vs night untreated, $p < 0.005$). These bumetanide-treated ZT17 mice thus behaved like the ZT5 experimental group.

We next performed the converse experiment, pre-treating five ZT5 mice for 40 min with VU0463271, to block KCC2. In 4 of these mice (80%), 4-AP treatment induced very intense epileptiform discharges, with sustained seizures (Fig. 6d, cyan), comparable to the level of pathological activity seen in the ZT17 mice. Thus, blocking NKCC1 appears to convert the ZT17 pattern to that of ZT5 mice, while blocking KCC2 converts the ZT5 to the ZT17 pattern.

To analyze these recordings further, we quantified the cumulative pathological activity in the four experimental groups (Fig. 6f, g). We found far higher levels of pathological activity at ZT17 (ZT5 vs ZT17, $p = 0.024$, Mann–Whitney test). Again, as with all the other electrophysiology and imaging assays, bumetanide treatment markedly altered the ZT17 pattern, pushing it towards that seen at ZT5 (ZT17 +/- bumetanide, $p = 0.008$; ZT5 vs ZT17+bumetanide, $p = 0.59$). We concluded therefore, that blocking NKCC1, using focally applied bumetanide, realigns the ZT17 cortical excitability to that at ZT5, as we had earlier shown with the direct visualization of the $[Cl^-]_i$ levels using LSSmClopHensor (Fig. 2c). Additionally, VU0463271 significantly

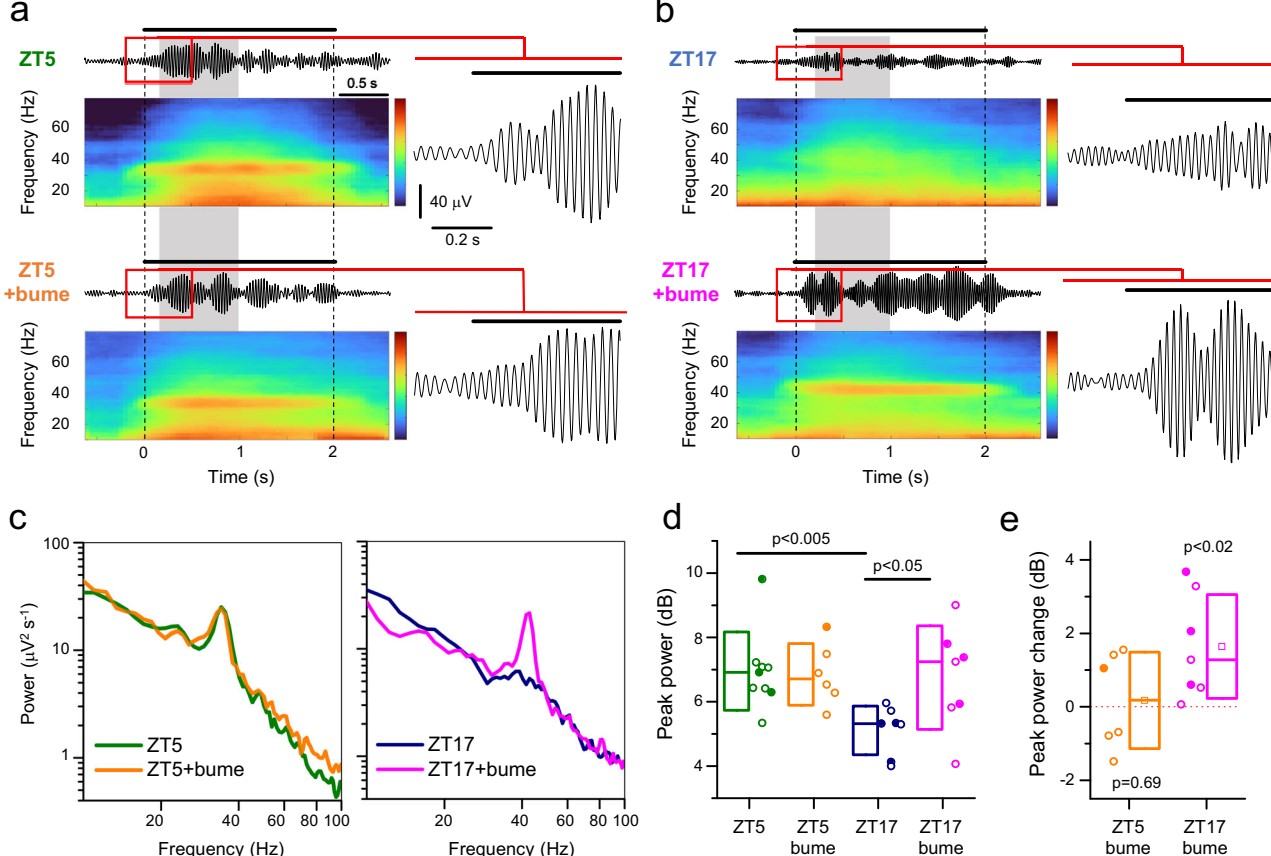

**Fig. 5 | Gamma-band synchronization differs between night and day in a chloride dependent way. a** Representative visually evoked responses to drifting gratings at ZT5 in a young adult mouse, initially recorded without bumetanide (top), and then repeated, in the same animal, 40–60 min after bumetanide (bume) application (bottom). The LFP is filtered in the 24–44 Hz band, around the peak of the gamma oscillation. **b** The equivalent data from a mouse recorded at ZT17. In each case, the LFP is presented with the spectrogram, and a zoomed in view of the start of the period of visual stimulation. Power spectra analyses were performed for the shaded epoch (0.2–1 s after the onset of the stimulus). Colormap limits: [−5,20] dB. Insets: broader view of the region in the red rectangle. Inset scalebar: 40 µV, 0.2 s. The LFP is filtered in the 31–51 Hz band. **c** An example of power spectra during the period of visual stimulation for individual mice recorded either at ZT5 (left) or ZT17 (right). Note the prominent peak in power in the gamma-band signal, at ZT5 and the lack of effect of bumetanide. Also note the absence of this peak at ZT17, but

its recovery following bumetanide application. **d** Box plots showing the prominence of the gamma-band, derived by subtracting the power spectra from a 1/f best fit of the 8–20 Hz and 80–110 Hz band and extrapolated over the entire frequency spectrum. Pooled data for all day and night-time experiments, showing a highly significant difference between the initial ZT5 versus ZT17 evoked responses ($n = 9$ ZT5, $n = 7$ ZT17). This day/night difference is lost after treatment with bumetanide ($n = 6$ ZT5, $n = 7$ ZT17). ZT5 vs ZT17+bume: $p = 2.1*10^{-3}$, Mann−Whitney tests, two-sided. ZT17 vs ZT17+bume: $p = 1.6*10^{-2}$, Wilcoxon Signed Ranks Test, two-sided. **e** Peak power increase after bumetanide superfusion, obtained from the paired data in (**d**). Two-sided Wilcoxon Signed Rank Test for whether the median differs significantly from zero (null change). Box plots in panel (**d**, **e**) represent the interquartile range, median (central line) and mean (small symbol). Source data are provided in the Source Data file.

increased the cumulative pathological activity at ZT5 (ZT5 vs ZT5 + VU0463271 $p < 0.05$, Mann−Whitney test) close to levels measured at ZT17 (ZT5 + VU0463271 vs ZT17 was not significantly different, Mann−Whitney test). These pharmacological experiments thus provide strong confirmatory evidence that the variations in these physiological (Figs. 4 and 5) and pathological (Fig. 6, Supplementary Fig. S11) assays do indeed arise from daily fluctuation in pyramidal [Cl⁻]$_i$.

## Discussion

We have shown here a previously unrecognized mechanism by which synaptic inhibition is modulated in mammalian neocortex during normal physiological daily rhythms, by varying neuronal intracellular [Cl⁻]. We provided a variety of corroborative imaging, biochemical, electrophysiological and pharmacological data, showing marked shifts in cortical excitability, from day (assayed at ZT5) to night (assayed at ZT17), aligned to a redistribution of Cl⁻ ions in and out of pyramidal neurons, in living mice. This result was replicated in different laboratories, despite subtle differences in the methodologies (Pisa used both

CD1 and C57B16 mice, transduced LSSmClopHensor by either in utero electroporation or viral vectors, and used Avertin anaesthesia; Newcastle used C57Bl6 mice, delivered LSSmClopHensor using viral vectors, and used urethane anaesthesia). Several measurements were made while animals were anaesthetized, which could affect the individual values, but importantly the same anaesthetic regimes were used for all time points through the day, so should not affect our interpretation regarding daily [Cl⁻]$_i$ modulation. The size and direction of [Cl⁻]$_i$ changes between day and night were almost identical in the two laboratories, indicating that the results are robust and independent of these methodological differences. We found that intracellular pH appeared almost constant across all times assayed, in contrast to a -9.5 mM increase in [Cl⁻]$_i$ from 8.9 mM at ZT5, when mice are typically asleep, to 18.4 mM at ZT17, when they are active. Interestingly, when the dark period was extended for 5 h into the following day, mice remained active yet [Cl⁻]$_i$ dropped to the same degree as seen in control mice (light on at ZT0). This indicates that the daytime chloride decline is uncoupled from the exposure to light at ZT0 and locomotor activity. Our experimental paradigm, which induces natural

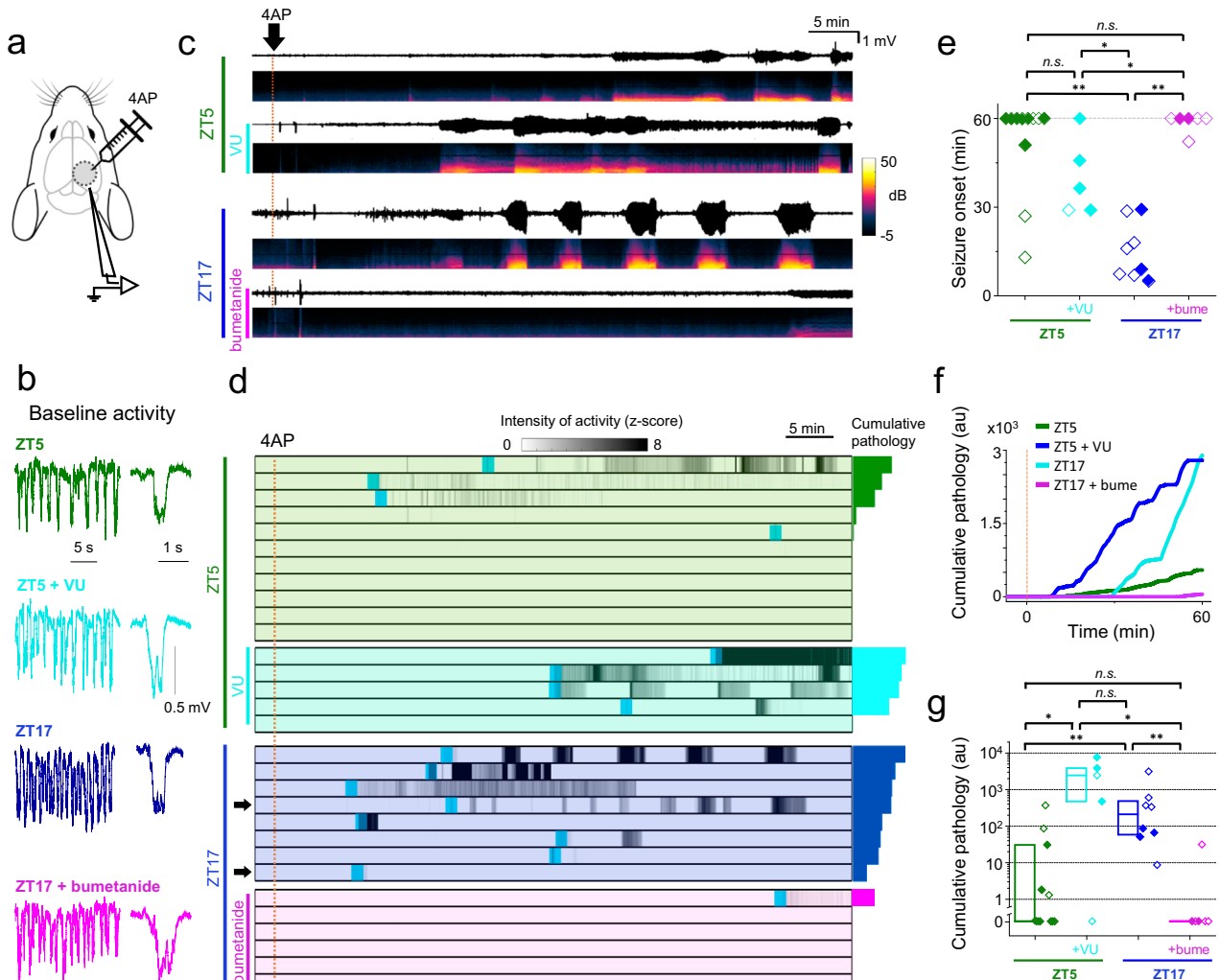

**Fig. 6 | Induced epileptic activity changes with the time of the day. a** Schematic showing the experimental arrangement. **b** Example LFP traces recorded prior to the application of 4-AP, showing the characteristic rhythmic activity pattern under urethane anaesthesia. In each case, we show a higher resolution of one of the cortical UP states (downward deflections on the LFP). **c** Example LFP traces, and corresponding spectrograms, for the first hour after an intracortical injection of 4-aminopyridine (4AP, dotted line), for the four different experimental groups: ZT5 without (green) / with VU0463271 (VU, cyan), and ZT17 without (blue) / with bumetanide (bume, magenta). Drugs were applied 40 min before of the 4AP injection. **d** Raster plots showing the temporal evolution of the z-score (grey-scale, see Methods and Supplementary Fig. S10 for details) for all mice included in the study. The light-blue shaded areas indicate the first seizures; these events are shown at higher resolution in Supplementary Fig. 11. The arrowheads in the ZT17 rasters indicate animals where saline was supplemented by 0.08% DMSO but no bumetanide ("vehicle controls"). **e** Latency to the first seizure. All animals which had no seizures within the hour are depicted as data points at 60 min. Open symbols indicate female mice. Statistical differences were tested using the Wilcoxon rank test (see Methods for the criteria used for burst identification). Two-sided Mann−Whitney tests. ZT5 vs ZT5 + VU: $p = 0.13$; ZT5 vs ZT17: $p = 1.3*10^{-3}$ (**); ZT5 vs ZT17 + bume: $p = 0.54$; ZT5 + VU vs ZT17: $p = 0.01$ (*); ZT5 + VU vs ZT17+bume: $p = 2.8*10^{-2}$ (*); ZT17 vs ZT17+bume: $p = 1.9*10^{-3}$ (**); no correction for multiple tests. **f** The average cumulative activity recorded for the three experimental groups within the first hour after the 4-AP injection. **g** The cumulative pathological activity (z-score) in the first hour following the 4AP injection. Box plots show the interquartile range, median (central line) and mean (small symbol). Two-sided Mann−Whitney tests. ZT5 vs ZT5 + VU: $p = 2.5*10^{-2}$ (*); ZT5 vs ZT17: $p = 5.0*10^{-3}$ (**); ZT5 vs ZT17+bume: $p = 0.32$; ZT5 + VU vs ZT17: $p = 0.21$; ZT5 + VU vs ZT17+bume: $p = 2.9*10^{-2}$ (*); ZT17 vs ZT17 + bume: $p = 3.0*10^{-3}$ (**); no correction for multiple tests; There was no statistical difference between sexes. Numbers of animals: ZT5: $N = 11$; ZT5 + VU: $N = 5$; ZT17: $N = 8$; ZT17+bumetanide: $N = 6$. Source data are provided as separate sheets in the Source Data file.

wakefulness without direct interaction with the mice, causes the mice to remain very active during this period of extended darkness, although it does not exclude the possibility that they sleep briefly at times. As such, the daily modulation of $[Cl^-]_i$ deviated from the behavioural activity during this period. In contrast, a recent study, in which the mice were sleep-deprived by experimental stimulation for the same period of time (ZT0-ZT4), found $[Cl^-]_i$ to be raised[60], suggesting that this paradigm of forced sleep-deprivation may differ from natural wakefulness, as has been reported elsewhere[61,62].

A pertinent question concerns the mechanism underpinning elevated pyramidal $[Cl^-]_i$ at night. Our biochemical measurements, and

pharmacological manipulations using either bumetanide or VU0463271 applied locally, both indicate that the redistribution is mediated by plasmalemmal cation-chloride co-transporters, meaning that the proximate source is the extracellular compartment. Extracellular changes may, in turn, be buffered, or even countered, by redistribution between other cell classes locally, or between the CNS and the rest of the body, across the choroid plexus. It may also be affected by redistribution of other ionic species ($Na^+$ and $K^+$ coupled through the electroneutral, chloride-cation cotransporters, NKCC1 and KCC2) as has been described in non-neuronal cells[22,63], or by changes in the membrane potential between different brain states,

which will additionally affect the driving force for ionic movement through GABA receptors ($V_m - V_{GABA}$). Circadian ionic redistribution observed in cardiomyocytes in vitro and cardiac tissue in vivo has been ascribed to isovolumetric circadian regulation of mTOR activity and crowding-associated compensatory transport of ions. Here, we believe that daily mTOR activation stimulates increased macromolecular crowding, with resultant increases in colloidal osmotic pressure counterbalanced by electroneutral export of $K^+$ and $Cl^-$ via KCC2[22,64]. Later in the circadian cycle when mTOR is inactive, crowding decreases and osmotic equilibrium is maintained by net ion import via NKCC cotransporters. This mechanism suggests a previously unsuspected link between circadian rhythms, protein homeostasis and baseline $[Cl^-]_i$ resulting in daily modulation of neuronal excitability, co-existing with faster ionic plasticity[21] that depends on overall neuronal activity.

It is helpful to consider how large the effect might be on the Nernst potential for $Cl^-$. If we were to assume a stable extracellular $[Cl^-]$, this doubling of $[Cl^-]_i$ equates to a median shift of $E_{Cl}$ from $-68.3$ mV to $-49.5$ mV, an 18.8 mV positive shift (for the purpose of this calculation we assume that the extracellular $[Cl^-]$ is 125 mM). In each animal, dozens of cells were imaged, and while there is inevitably some experimental error in these values, the interquartile ranges in each animal were relatively tight (8–18 mM ranges). Importantly, the accuracy of the median values, which is what we focus on, was verified directly in a previous study in brain slices[65], where the estimates of $[Cl^-]_i$ derived using LSSmClopHensor imaging tightly matched those derived from perforated patch recordings.

These measurements were made from supragranular pyramidal cells, which clearly represent only a limited sampling of the variety of cell classes found in the central nervous system, but it is noteworthy for two important reasons. First, we demonstrate the physiological modulation of baseline $[Cl^-]_i$ in vivo, in cortical neurons. Interestingly, patch clamp recordings of neocortical L2/3 pyramidal cells, in brain slices prepared from adult mice at different times of day, also showed increased mini-inhibitory postsynaptic current frequency and an increased cumulative IPSC conductance (measured over 1 s periods) during the day, that was suggested to reflect increased activation of endocannabinoid receptors[66]. That study, however, was performed using whole-cell patch clamp configuration, which would normalize the $[Cl^-]_i$ across all cells and therefore would not detect the mechanism we describe here. Indeed, a more recent study, performed using the perforated patch configuration, revealed changes of $Cl^-$ reversal during the day[60], consistent with the data we present here. Second, the pyramidal cell modulation is out of phase with the oscillation in the majority of SCN neurons[51], although there is heterogeneity in the phase of oscillation across the SCN[67,68]. This suggests that the pattern of $[Cl^-]_i$ modulation may also differ between cortical cell classes, which is the topic of ongoing investigation. As such, the physiological and biochemical assays do not present the full picture, because neither examine changes at the resolution of single cells, or cell classes.

The dynamics of the $[Cl^-]_i$ are entirely consistent with changes in the phosphorylation state and expression levels of the key chloride-cation cotransporters (Fig. 3), suggesting a causal link. The imaging data and the biochemical assays were further corroborated by two functional assays showing pronounced day-to-night differences in cortical state: one physiological, where we showed an alteration of surround inhibitory effects in primary visual cortex (Figs. 4, 5)[59], and the other pathophysiological, with a greatly reduced latency to induced seizures (Fig. 6), indicative of a weakened inhibitory restraint[69,70], at ZT17. Of particular importance is the fact that for all three data sets—the LSSmClopHensor imaging, the surround inhibition, and the latency to seizures—the ZT17 pattern was converted to the ZT5 pattern by focal application of bumetanide, blocking NKCC1, and thereby reducing $[Cl^-]_i$. This strongly suggests that the functional effects we document are all mediated locally, and we may discount more systemic influences, such as drugs acting at the choroid plexus,

or for the visual assays, possible confounding effects arising from day-to-night changes either in the retina or LGN. Of relevance also is that Veit and colleagues convincingly showed that the visually evoked oscillations are not seen in earlier visual structures, but rather that they are mediated locally through the somatostatin interneuron population[59].

The finding that fluctuations in pyramidal $[Cl^-]_i$ were also associated with large changes in epileptic susceptibility at different times of the day, is consistent with the well-recognized clinical phenomenon of circadian clustering of seizures[2,3,71,72], although note that the phase relation varies widely between individual patients. Given the direct coupling of the changes in $[Cl^-]_i$ to the biochemical state of neurons, and the well-established role of synaptic inhibition in shaping all manner of neuronal activity[73–79], we suggest that the changes we describe here, could constitute an important determinant of the distinct neuronal activity patterns, contributing to the cycle of brain states through the day. There are, of course, other neuronal daily fluctuations, including changes in firing rates[7–9,80], in the number, strength and structure of synapses[10–13], metabolic state[16], gene expression and protein phosphorylation[17,18]; the relative importance of these different factors, and the degree to which they are coordinated, or might offset each other, will be an important topic of future research.

## Methods

### Animals and housing conditions
All animal work was performed according to the guidelines of the Home Office UK and animals (Scientific Procedures) Act 1986, and to the protocol 211/2020-PR approved by the Ministero della Salute (Italy). Mice were kept in a steady 12-h light / dark cycle from birth, with light being turned on at 7:00 and turned off at 19:00. Food and water were provided ad libitum. Chloride imaging was performed on both CD1 and C57 mice, biochemistry analysis and electrophysiology was performed on C57 mice only. We examined both female and male mice, in approximately equal proportions.

### Production and purification of the AAV8-EF1a-DIO-LSSmClopHensor vector
The virus was produced by the triple transfection method, using 293AAV cells (Cell Biolabs). 293AAV cells were expanded in D-MEM supplemented with 10% FBS, L-glutamine and Pen-Strep, to then transfect $40 \times 150$ mm plates. For each 150 mm plate, $1.5 \times 10^7$ cells were plated the day before the PEI-mediated transfection (Polysciences). For each plate were used 30 μg of pAdDeltaF6 helper plasmid (Addgene plasmid # 112867), 15 μg of pAAV2-8 Rep-Cap plasmid (Addgene plasmid # 112864), and 15 μg of pAAV-EF1a-DIO-LSSmChlopHensor. Sixteen hours after transfection, medium was replaced with D-MEM supplemented with 5% FBS, L-glutamine and Pen-Strep. 72 h after transfection, cells were scraped and collected by centrifugation.

The virus was purified as described previously[81]. Briefly, the cell pellet was resuspended in sodium citrate pH 8.05 (100 mM) using 1.44 ml of it + 2.25 of $H_2O$ per ml of cell pellet. Cells were then sonicated, and 1 M magnesium chloride was added to a final concentration of 1.6 mM. 200 U of benzonase (Sigma-Aldrich) per ml of cell pellet were added and incubated 1 h at 37 °C. After the benzonase reaction, 3.06 ml of citric acid (100 mM) + 2.25 ml of $H_2O$ per ml of cell pellet were added to clarify the crude extract by protein flocculation. The extract was then centrifuged at 4500 g at RT for 10 min. The supernatant was recovered and filtered through a 0.22 μm PES filter (Millipore), and directly applied to a HiPrep SP HP 16/10 column (Cytiva) using an AKTA start (Cytiva) connected to a fraction collector Frac30 (Cytiva).

The virus was diafiltrated (1:160000) and concentrated (to final volume of ~1 ml) using Protein Concentrators PES, 100 K MWCO

**Table 1 | Details of the antibodies used for the immunolabelling experiments**

| ANTIBODIES | CONCENTRATION |
|---|---|
| GFAP: ab4674 (Abcam), Chicken | 1:1000 |
| GFP: ab290 (Abcam), Rabbit | 1:2000 |
| GFP: ab13970 (Abcam), Chicken | 1:1000 |
| CamKII: ab52476 (Abcam), Rabbit | 1:500 |
| 488-anti-rabbit: A21206 (Invitrogen), Donkey | 1:500 |
| 568-anti-rabbit: A10042 (Invitrogen), Donkey | 1:500 |
| 488-anti-chicken: 703-545-155 (Jackson Immuno), Donkey | 1:500 |
| 647-anti-chicken: 703-605-155(Jackson Immuno), Donkey | 1:500 |

(Thermo Scientific, Pierce). The final buffer for the virus was: dPBS + 5% Glycerol + 0.001 Pluronic F-68 (Invitrogen). The AAV titre ($1.71 \times 10^{13}$ GC/ml) was determined by qPCR using a standard curve of linearised pAAV-EF1a-DIO-LSSmChloPhensor plasmid as reference.

## LSSmClopHensor expression

Expression of LSSmClopHensor was achieved either by *in utero* electroporation or by viral vectors. *In utero* electroporation was performed in CD1 mice at embryonic day 15.5 with a custom-made triple electrode to transfect neuronal progenitors of layer 2/3 pyramidal neurons of the visual cortex. Details have been presented elsewhere[82,83] Transfection using the AAV-LSSmClopHensor viral vectors was performed on neonatal mouse pups (postnatal day 0–1) expressing Cre-recombinase under the Emx1 promoter, on a C57Bl6j background (Jackson lab #5628), following a previously described protocol[33]. In short, pups were anaesthetized using isoflurane (2% in $O_2$ delivered at 0.8 L/min). The scalp and skull were perforated using a 23 G needle, approximately midway between bregma and lambda, and roughly 1 mm lateral to the midline, on the left side. A Hamilton Nanofil syringe needle (36 G, World Precision Instruments) was then guided through the hole, to a depth of 1.7 mm, aiming for the lateral ventricles, and 500 nl of AAV8 EF1a DIO LSSmClopHensor viral vector was delivered at that depth, and further boluses also delivered, at 2 min intervals, also at 1.4, 1.1 and 0.8 mm depth. The needle was left in situ for at least 3 min after the final bolus injection. Mice were allowed to recover fully, while being kept warm, before being returned to the mother.

## Immunolabeling of LSSmClopHensor in fixed brain tissue

After urethane anaesthesia mice were intracardially perfused with 4% PFA in PBS (Santa Cruz). The brain was dissected and stored in 4% PFA. 40 μm coronal sections were obtained using a freezing microtome and the immunohistochemistry was performed as previously described[84]. Briefly, following antigen retrieval (10 mM Na-citrate buffer pH6), slices were incubated in blocking buffer (10% donkey serum, 0.05% TritonX in PBS). Slices were then incubated overnight in blocking buffer containing the primary antibodies. After three washes in PBS, slices were incubated in blocking buffer containing secondary antibodies. After three washes in PBS the slices were mounted on gelatin-coated glass slides using Fluoromount-G mounting media (Invitrogen). All the antibodies' details are reported in Table 1. The GFP antibody binds to the modified eYFP component of LSSmClopHensor, to provide a boosted fluorescent signal in fixed tissue.

Confocal images were acquired on a Zeiss LSSM880 upright microscope using a water immersion 40x lens: W Plan-Apochromat 40x/1.0 DIC VIS-IR M27 (Newcastle University, Bioimaging unit). All images shown were acquired in the superficial layers of the cortex, at a depth compatible with the 2photon images acquisition (not more than 250–300 μm from the pial surface). At least 3 slices per condition were examined. Image analysis was performed using Fiji.

## LSSmClopHensor imaging

All imaging experiments were performed as acute terminal experiments on young mice (age 1–4 months) expressing LSSmClopHensor. Preparatory surgery followed two slightly different protocols in the two centres. In Pisa, mice were anaesthetized with 2,2,2-tribromoethanol (Avertin) (i.p. 0.02 ml/g body weight) whereas in Newcastle, urethane anaesthesia was used (i.p. 1.6 g/kg) and supplemented with low level of isoflurane inhalation (5% in $O_2$ for induction, and then lowered to <1% in $O_2$ at 0.6–1 L/min, for maintenance during the craniotomy, but discontinued for >20 min, prior to imaging). In both cases, a 4 mm craniotomy was performed over the occipital cortex and covered with a 4 mm-coverslip. In some experiments, the coverslip was left partially open on a side to allow for topical application of bumetanide (55 μM in saline and 0.08% DMSO) or VU0463271 (10 μM in saline and 0.1% DMSO), applied to the surface of the brain, as described previously[28]. Imaging was performed immediately after surgery.

All imaging sessions were performed in identical fashion, at four different times during the day: start of the light period (ZT2, 09.00 h), midday (ZT5, 12.00 h), start of the dark period (ZT12, 19.00 h) and midnight (ZT17, 23–24.00 h). Once mice were fully anaesthetized, they were positioned underneath a 2-photon microscope. Images were acquired at five different excitation wavelengths (800 / 830 / 860 / 910 / 960 nm−some of the 800 nm data sets were only fractionally above noise levels and so were not used for analyses) and collected through green and red emission filters (527/70 nm and 607/70 nm respectively, BrightLine). Post hoc analysis of the images was performed using a graphical user interface (GUI), implemented in Matlab (Mathworks, MA, USA), that automated finding cell somata, to collect red/green fluorescence ratios at the different wavelengths, and compute intracellular pH and [Cl⁻] values for each cell, based upon earlier calibration experiments performed using ionophore-permeabilized HEK cell and neuronal cultures[28].

The two centres used two different imaging setups. Pisa: Bruker Ultima with a Coherent Ultra II laser and an Olympus 20X, NA 1.05 objective (water immersion). Newcastle: Bergamo II 2-photon microscope, with a MaiTai laser (SpectraPhysics) and a Nikon 16X, NA 0.8 (water immersion). Of note, both microscopes were equipped with identical filters on the emission pathway.

## Visual responses recording

The implant for the head fixed mice was prepared under Avertin anaesthesia (20 μL/g) and after application of local anaesthetic (Lidocaine Hydrochloride 1% gel; Molteni Farmaceutici). The head of the mouse was gently fixed in a stereotactic frame, skin was removed, and the skull was cleaned from membranes and dried. A small hole (0.2 mm Ø) was drilled in the skull in the right hemisphere in correspondence of V1 (0.0 mm anteroposterior and 2.5 mm lateral to the lambda suture). A tungsten microelectrode (FHC, outside diameter 200 μm) controlled by a motorized micromanipulator (Sutter Instrument MP C–200), was inserted in the hole at a depth of 250–300 μm and fixed with dental cement (Vertex Dental), together with a metal frame for the purpose of head restraint. After surgery, the mouse rested for 4 days and was subsequently trained for 3 days (2 training sessions per day) to familiarize with the experimental setup. Visual stimuli consisted of full-contrast drifting square-wave gratings at 0.04 cycles per degree and 2 cycles s⁻¹ moving horizontally from right to left and with a mean luminance of 10 cd/m². Each recording consisted of at least 15 repeated trials. These stimuli were generated using the Matlab toolbox Psychophysics Toolbox (Brainard, 1997). Each trial consisted of 2 s of stimulus presentation followed by 2 s of equiluminant uniform grey. Stimuli were presented only to the contralateral eye, at a distance of 15 cm, via a PC monitor and the geometry of the grating was corrected to compensate for the distortion due to the proximity of the stimulus to the eye. The visual stimuli and responses were synchronized using a phototransistor

(LPT 80 A, Osram) connected to a microcontroller, and recorded through a parallel channel to the LFP recordings.

Electrophysiology on anaesthetized C57 mice was performed under urethane anaesthesia (i.p. 1.6 g/kg) and bumetanide was applied topically as described for the imaging experiments. The mouse was placed in a stereotaxic head-holder and a small craniotomy was performed to reveal the cortex. Recordings were performed from layer 2/3 of the primary visual cortex (0.0 mm anteroposterior and 3.0 mm lateral to the lambda suture) by means of a 2 MOhm glass microelectrode with a AgCl electrode.

## Seizure susceptibility

In vivo electrophysiological experiments were performed on acutely prepared C57 mice, under urethane anaesthesia (i.p. 1.6 g/kg). The mouse was placed in a stereotaxic head-holder and a small craniotomy was performed to reveal the cortex. Recording was performed either using 2 MOhm glass microelectrodes or a multi-electrode arrays (MEA, Neuronexus−A4x4-4mm-200-200-1250) and digitised at 10 kHz (NPI EXT-02F amplifier, acquired either with a 16-bit AD board USB6251 from National Instruments or with the Neuronexus SmartBox Pro, Plexon AC amplifier and stored using the SortClient software). In both cases, electrodes were introduced into the cortex to a depth of 0.3 mm, and approximately positioned within the primary somato-sensory cortex[85]. The cortex was kept moist with warmed saline throughout recording sessions. LFP signals were amplified 500 times (EXT − 02 F amplifiers; NPI electronic), band pass filtered (0.1 − 1000 Hz), cleared from 50 Hz electrical noise (Hum Bug Noise Eliminator; Quest Scientific), and finally sampled at 10 kHz.

Epileptic activity was induced acutely by a single bolus injection of 500 nl of 15 mM 4-aminopyridine in saline (total amount delivered 7.5 nmol), injected at 1 μl/min via a NanoFil needle (36 G, World Precision Instruments; 30 s injection), roughly 1 mm caudal to the recording site (approximately into the anterior end of primary visual cortex). In all the animals, saline was applied directly to the surface of the brain after the craniotomy was made, to keep it moist. In a subset of experiments, this saline solution was supplemented with either 55 μM bumetanide (in 0.08% DMSO and saline) or VU0463271 (10 μM in saline and 0.1% DMSO, as described for the preceding experiments) for ~1 h before 4-AP injections. To test for effects of this trace amount of DMSO, in two experiments (arrowheads in Fig. 5d), saline was applied with just 0.08% DMSO.

## Electrophysiology analysis

For the visual stimulation experiments, local field potentials were recorded in primary visual cortex, and short epochs aligned to the stimulus onset, were analysed using a custom-written code in Matlab (all code is available at https://github.com/GabNar/Pracucci_Graham_Alberio_Nardi_et_al_2023). Power spectra were computed in 800ms-long windows during the visual stimulation (from 200 ms to 1 s after the stimulus onset) or during the presentation of a grey screen equiluminant to the square wave grating (baseline). For each animal, the power spectra of 30−60 trials were averaged. The amplitude of the gamma-band peak was obtained by subtracting the power spectra from a 1/f best fit of the 8−20 Hz and 80−110 Hz band and extrapolated over the entire frequency spectrum. In 2 mice recorded during the day and in 1 mouse recorded during the night, the status of the animal deteriorated right before and then after topical bumetanide application, as electrophysiological responses to visual stimuli drastically diminished; for this reason, the post-bumetanide condition is missing for those mice and those traces were excluded. The data depicted in Figs. 4 and 5 have been recorded with electrodes of different dimensions and impedance and in very different experimental settings, thus are characterized by different S/N ratios. For this reason, the distribution of the spectral power has been visualized and quantified differently (see Supplementary Fig. S8).

For the seizure susceptibility experiments, extracellular field recordings were analysed to detect pathological discharges using a custom-written code in Matlab. In brief, this used a frequency-domain analysis to detect periods when the LFP power exceeded baseline activity by a designated threshold (Supplementary Fig. S10). The spectrogram (bin width 10 s, with 1 s time shift) of the baseline (the period preceding 4AP injection) was computed and we obtained mean and standard deviation of the spectral power for frequencies between 8 and 100 Hz. The spectrogram of the entire recording was normalized to the baseline mean and standard deviation for each frequency bandwidth individually. To avoid contamination with mains noise, we omitted frequencies between 49−51 Hz. This process generated a Z-score spectrogram ($Z = (x-mean)/s.d.$), in which each element of the Z-score matrix represented the excess power at a particular frequency, for each time bin. Using a threshold of 4 standard deviations above the baseline mean, we identified all time bins for which any frequency exceeded this threshold. The power integrated in each time bin was summed cumulatively, to estimate the rate and acceleration of pathological activity within the brain. Continuous epochs of supra-threshold bins were designated to be single "events". Seizures were identified as continual large amplitude rhythmic discharges lasting >15 s. In fact, most seizures lasted for many tens of seconds, before transitioning into a period of post-ictal suppression.

## KCC2/NKCC1 analysis

**Animals and sample preparation.** Brain tissue was collected from the same mouse colony that was used also for imaging and electrophysiology experiments and housed and monitored prior to sacrifice as described earlier. Tissue collection was made at either midday (ZT5, 12.00 h) or midnight (ZT17, 24.00 h). Mice were killed by cervical dislocation, and the brains removed into ice-cold saline, and a rapid dissection was made to isolate right and left neocortical and hippocampal brain tissue samples, which were immediately flash-frozen in liquid nitrogen, and stored subsequently at −80⁰. All biochemical assays and analysis were performed blind to the ZT5/ ZT17 grouping. Protein samples were prepared in NuPAGE™ LDS Sample Buffer (reducing conditions) and were incubated in the same sample buffer at 75 °C for 10 min, prior to loading on the gel.

**Antibodies.** The primary antibodies used are all as described in previous publications, where the validation of the labelling is presented in detail[56–58]: anti-KCC2 phospho-Ser940 (Thermo Fisher Scientific, cat #PA5-95678), anti-(neuronal)-β-Tubulin III (Sigma-Aldrich, cat #T8578), anti-KCC2 total (University of Dundee, S700C), anti-KCC2 phospho-Thr1007 (University of Dundee, S961C), anti-NKCC1 total antibody (University of Dundee, S022D), anti-NKCC1 & NKCC1 phospho-Thr203/Thr207/Thr212 antibody (University of Dundee, S763B). Horseradish peroxidase-coupled (HRP) secondary antibodies used for immunoblotting were from Pierce. IgG used in control immunoprecipitation experiments was affinity-purified from pre-immune serum using Protein G-Sepharose.

**Buffer for western blots.** Buffer A contained 50 mM Tris/HCl, pH 7.5 and 0.1 mM EGTA. Lysis buffer was 50 mM Tris/HCl, pH 7.5, 1 mM EGTA, 1 mM EDTA, 50 mM sodium fluoride, 5 mM sodium pyrophosphate, 1 mM sodium orthovanadate, 1% (w/v) NP40, 270 mM sucrose, 0.1% (v/v) 2-mercaptoethanol, and protease inhibitors (complete protease inhibitor cocktail tablets, Roche, 1 tablet per 50 mL). TBS-Tween buffer (TTBS) was Tris/HCl, pH 7.5, 150 mM NaCl and 0.2% (v/v) Tween-20. SDS sample buffer was 1X NuPAGE™ LDS Sample Buffer (NP0007, Invitrogen™), containing 1% (v/v) 2-mercaptoethanol. Protein concentrations were determined following centrifugation of the lysate at 16,000 x g at 4 °C for 20 min using the Pierce™ Coomassie (Bradford) Protein Assay Kit (23200, Thermo Scientific) with bovine serum albumin as the standard.

**Surface biotinylation in cerebral cortex.** Biotinylation studies were performed as previously described[86] with modifications. Mice were killed by cervical dislocation and the brain rapidly extracted and placed in a cold (~4 °C), oxygenated sucrose-based solution, comprising 189 mM sucrose, 10 mM d-glucose, 26 mM NaHCO$_3$, 3 mM KCl, 5 mM MgSO$_4$, 0.1 mM CaCl$_2$ and 1.25 mM NaH$_2$PO$_4$. Horizontal sections (400 μm) were made from 5–7-week-old wild-type animals (C57Bl/6j, Janvier) using a vibrating blade microtome (VT1200, Leica Microsystems, Wetzlar, Germany) in N-methyl-D-glucamine (NMDG) based cutting solution containing 93 mM (NMDG) titrated to pH 7.4 with HCl, 2.5 mM KCl, 1.2 mM NaH$_2$PO$_4$, 30 mM NaHCO$_3$, 20 mM HEPES, 25 mM glucose, 5 mM ascorbic acid, 3 mM sodium pyruvate, 10 mM MgCl$_2$, 0.5 mM CaCl$_2$, saturated with 95% O$_2$/5% CO$_2$, pH 7.4, 300 mOsm. After cutting, the slices were immediately removed to a holding chamber containing oxygenated (95% O$_2$–5% CO2) artificial cerebrospinal fluid (aCSF) comprising: 124 mM NaCl, 3 mM KCl, 24 mM NaHCO$_3$, 1 mM MgSO$_4$, 10 mM d-glucose and 1.2 mM CaCl$_2$. The slices were gradually warmed to ~37 °C (for 30 min) and then maintained at room temperature (~20 °C, for at least another 30 min) until ready for use. The slices were transferred into bubbled ice-cold aCSF containing 1 mg/ml EZ-Link™ Sulfo-NHS-Biotin (21217, Thermo Scientific) with gentle rotation for 45 min at 4 °C. Excess biotin was quenched using 1 M glycine in ice-cold aCSF for 10 min, and then the slices were rinsed once in ice-cold aCSF and snap frozen on dry ice. The cerebral cortex was micro-dissected and immediately lysed and homogenized in lysis buffer. The samples were centrifuged at 16,000 x g at 4 °C for 20 min, the supernatant was collected, and protein content was determined using a Pierce™ Coomassie (Bradford) Protein Assay Kit (23200, Thermo Scientific). 40 μg of protein was loaded onto 100 μl of 50% slurry of Pierce™ NeutrAvidin™ UltraLink™ Resin (53150, Thermo Scientific), made up to a total volume of 400 μl in lysis buffer and rotated for 2 h at 4 °C. The beads were recuperated by centrifugation and thoroughly washed four times in lysis buffer, and after the last wash the beads were incubated in ×2 SDS sample buffer containing 10% β-mercaptoethanol at 37 °C for 1 h. The protein samples were run on precast Bis-Tris gels (NuPageTM 4–12% gradient gels, NP0322, Invitrogen™) and immunoblotting was performed. Analysis was performed using ImageJ by normalizing the amount of surface KCC2 to the amount of β-tubulin in the non-biotinylated fraction.

**Immunoblot and phospho-antibody immunoprecipitation analyses.** Cerebral cortex lysates were subjected to immunoblot and immunoprecipitation as previously described[56]. Cerebral cortex lysates (20 μg) were boiled at 75 °C in sample buffer for 10 min, resolved by 7.5% sodium dodecyl sulfate polyacrylamide-gel electrophoresis and electrotransferred onto a polyvinylidene difluoride membrane. Membranes were incubated for 30 min with TBST (Tris-buffered saline, 0.05% Tween-20) containing 5% (w/v) skim milk. Blots were then washed three times with TBST and incubated for 1 h at room temperature with secondary HRP-conjugated antibodies diluted 5000-fold in 5% (w/v) skim milk in TBST. After repeating the washing steps, signals were detected with enhanced chemiluminescence reagent. Immunoblots were developed using ChemiDoc™ Imaging Systems (Bio-Rad, Feldkirchen). Figures were generated using Photoshop/Illustrator (Adobe). The relative intensities of immunoblot bands were determined by densitometry with ImageJ software. Calculation of intensity ratios[87] is based on (phospho-dimeric KCC2 + phospho-monomeric KCC2)/(total dimeric KCC2 + total monomeric KCC2), (total dimeric KCC2 + total monomeric KCC2)/β-tubulin, NKCC1 pThr203,207,212 /NKCC1, NKCC1/β-tubulin, as reported previously[56]. Data were expressed as means SEM. Statistical significance was determined by Wilcoxon–Mann–Whitney sum test and adjusted by Bonferroni correction (GraphPad Prism 7.0, San Diego, CA, USA).

KCCs phosphorylated at the KCC2 Thr[1007] equivalent residue were immunoprecipitated from clarified clarified cerebral cortex lysates (centrifuged at 16,000 x g at 4 °C for 20 min) using phosphorylation site-specific antibody coupled to protein G–Sepharose as previously described[56]. The phosphorylation site-specific antibody was coupled with protein-G–Sepharose at a ratio of 1 mg of antibody per 1 mL of beads in the presence of 20 μg/mL of lysate to which the corresponding non-phosphorylated peptide had been added. Two mg of clarified cell lysate were incubated with 15 μg of antibody conjugated to 15 μL of protein-G–Sepharose for 2 h at 4 °C with gentle agitation. Beads were washed three times with 1 mL of lysis buffer containing 150 mM NaCl and twice with 1 mL of buffer A. Bound proteins were eluted with 1X LDS sample buffer.

## Data availability
All data will be made available upon request. Source data are provided with this paper.

## Code availability
All custom code has been written in ImageJ and MatLab and it is available at https://github.com/GabNar/Pracucci_Graham_Alberio_Nardi_et_al_2023.

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

## Acknowledgements

We thank Melissa Santi and Simone Giubbolini for help during the calibration of the 2-photon microscope, Andy Jackson, Claudia Lodovichi, Laura Baroncelli, Giorgio Carmignoto, Paola Tognini and Mike Hausser for helpful discussions during the project. We are grateful to Francesca Biondi for animal care (Pisa) and to the support staff in the animal facility at Newcastle University. Illustrations in Fig. 2f and Supplementary Fig. S8 were created with Biorender.com. This work was supported by: Telethon grant GGP19281 (G.M.R.); Regione Toscana project DECODE-EE (G.M.R); Royal Society UK grant IEC\NSFC\201094 (J.Z.); BBSRC BB/P019854/1 (A.T.); MRC MR/R005427/1 (A.T.); Epilepsy Research UK (Celine Newman Bursary). A.T. and G.M.R. are supported by a joint Royal Society/CNR grant. L.A. holds a Newcastle University Faculty Fellowship; R.T.G. held a Newcastle University Faculty PhD studentship; G.N., V.P. and G.P. held Scuola Normale Superiore PhD studentships.

## Author contributions

Design of the project: G.M.R., A.J.T. In vivo imaging: R.T.G., L.A., A.J.T., O.C., E.P., S.L., G.P., G.N. Electrophysiology: E.P., G.N., V.P., G.P., R.T.G., L.A., S.L. Viral vector production: L.S., L.A. In utero transfection: V.P., E.P., O.C., S.L., G.N. Analyses: R.T.G., L.A., D.W., A.J.T., G.P., G.N. Biochemical analyses: J.Z. Computer coding: E.P., R.T.G., G.N., A.J.T., G.M.R. Figure preparation: G.M.R., R.T.G., A.J.T., G.N., G.P., E.P. Writing manuscript, A.J.T., G.M.R. All authors reviewed the final manuscript.

## Competing interests

The authors declare no competing interests.
