## [Peer Review File · Nature Communications]

Daily rhythm in cortical chloride homeostasis underpins functional changes in visual cortex excitabilityReviewers' Comments:

Reviewer #1:

Remarks to the Author:

I would like to congratulate the authors on this excellent contribution, that I believe will become a citation classic, since it dramatically extends understanding of how neuronal chloride homeostasis, and its temporal variation, impacts upon neuronal electrophysiology in the adult mammalian brain. Using a very sophisticated fluorescent probe for neuronal chloride levels, previously developed by Ratto, they show daily variation in pyramidal neurons in living mice being higher during the night/active phase. This extends previous findings (from SCN neurons by Jen Evan's and Doug McMahon's labs, and assorted non-neuronal cells by Stangherlin/O'Neill labs) into the cortical neurons of adult brain. In itself, this is a major findings, since daily variation in the chloride reversal potential would be expected to lead to daily variation in the potency and direction of GABAergic neurotransmission. They go on to demonstrate that variation in Chloride levels is consistent with differential daily variation in NKCC/KCC activity.

The authors then test two functional predictions that the time of day-dependent variation in 1) gamma-band synchronisation by visual stimulation in visual cortex and 2) susceptibility to epilepsy can be selectively blocked by bumetanide at night (inhibits NKCC mediated chloride efflux). Overall this is excellent hypothesis driven neuroscience that can be expected to have a broad impact, because it raises as many follow-up questions. I have a few comments, as follows:

Major points

1. Although it is technically correct that these experiments were performed under diurnal conditions, many (potentially) interested readers will not know the word or may be confused by knowledge that the normal behaviour of a mouse is nocturnal. To more clearly communicate their findings to a wide audience, I would strongly encourage the authors to use "daily" in place of "diurnal", both in the title and throughout, since this is unambiguous and more accessible.
2. There is no clear justification for the animal numbers used in each experiment or the number of neurons recorded from each mouse. Neither is there any clear explanation of whether there was consistently 50:50 male:female mice, and if not why not. This is not a fatal problem but must be addressed in the methods section.
3. The text and data presentation explaining Fig 5/6 need to be improved. The text states that two different experimental paradigms were employed: injection of 4-AP into neocortex vs injection of 4-AP into neocortex followed by direct application of bumetanide. There does not appear to be a "vehicle" control for the latter. This is a concern because, in principle, that the act of applying the same solution without bumetanide might also affect seizure susceptibility during the night. I really would expect this control to be performed in order to have confidence in the effect of bumetanide.

Minor points:

Interaction with anaesthesia?

It would be helpful to have a cartoon schematic to explain the experimental setup and data processing that allows one to get to Fig3c. I know this is partly explained in the text and partly in the methods section, but it would help the reader enormously to see the workflow diagrammatically.

It was not immediately clear to be how the analysis presented in Fig 4b/c differs from that in Fig3c. Please can this be explained more clearly of the method of analysis harmonised?

Line 40 - adult mouse neocortex not mice

Line 54 - "Basic animal research has further documented differences in the level of neuronal..." ... presumably means "Basic animal research has further documented time -of-day differences in the level of neuronal..."

Line 95/129 and elsewhere - "the 24hr period" should be "day/night cycle"?

Line 105 - worth justifying choice of neuron/location?

Line 115 - daily rest/activity cycle instead of diurnal cycle

Line 131 - mouse's not mice's

Line 484 - mouse pups

Line 209 - means Fig 2c?

Line 237-9 - sentence ends with a reference to Fig5b, but isn't clear how the data shown in 5b relates to preceding text since 10 min baseline is not shown, neither are the characteristic UP states that are referred to. Please clarify.

Line 285 - NKCC and KCC are meant to be electroneutral transporters so from the context I presume the authors mean "accompanied by daily redistribution of cation counterions" rather than the rather vague "affected by circadian redistribution of other ionic species" ...if the same type of process described by Stangherlin is operating here?

Line 302 - should also refer to prior work on circadian rhythms in SCN neuronal chloride levels by Jen Evans and Doug McMahon's lab.

Discussion - a sentence or two about what might generate these daily changes in Cl levels would not go amiss.

Fig 2a - SDS-PAGE was run under reducing conditions, how are these dimers occurring/being detected?

Fig 2b - might be clearer to use pm-KCC2 and all-KCC2 or similar, and do the same for NKCC1 for consistency. Either way, which was used for the denominator in the ratios in top right and middle left

Fig 3a/b & 4a. Heat maps are missing a y-axis label

Fig 4b-e. Please report number of animals

Fig 6. Please show all traces in a supplementary figure, not just those with an ictal event.

Reviewer #2:

Remarks to the Author:

Nature Communications manuscript NCOMMS-22-46047-T

In the manuscript Diurnal rhythm in cortical chloride homeostasis underpins functional changes in visual cortex excitability the authors present interesting new data showing that Cl in cortical neurons differs significantly at different times of the day. The time dependent Cl concentration is correlated with whole brain phosphorylation levels of NKCC1 and membrane expression of KCC2. Furthermore,

application of the NKCC1 targeting drug bumetanide affects visual processing and induced epileptic activity. While the basic finding of this study is highly relevant and new, the underlying mechanism which is presented is questionable based on the data presented.

Major comments:

1. Western blot analysis in Figure 2 was performed on whole brain tissue without discussing which cell types might contribute to this result. While there is a difference in membrane expression of KCC2, NKCC1 only differs in phosphorylation levels. Please discuss in more detail how different phosphorylation can affect the activity of NKCC1. The amount of membrane bound NKCC1 has not been determined.
2. The authors aim to manipulate neuronal Cl using bumetanide. This needs to be shown by imaging neuronal Cl with and without bumetanide. In the following experiments bumetanide was applied for different amounts of time, please show for the different duration (40 and 60 min) how neuronal Cl is affected by bumetanide.
3. The major part of the presented data was performed under two different kind of anesthesia, which are both known to affect anion channel in different ways. It is crucial to highlight which anesthetic was used in which experiment, compare neuronal Cl- under both anesthetics and discuss how those might affect the presented data.

Minor comments:

1. Two different mouse lines were used. Which line was used for which experiment, are Cl dynamics and expression pattern for NKCC1 and KCC2 the same?
2. Expression of the Cl sensor ClopHensor was obtained by two different ways, in utero electroporation and virus injection. IHC staining showing expression pattern and cell type specificity are needed.
3. The activity pattern of the mice used for Cl imaging was tracked, does activity correlate with the Cl level determined? Can different activity levels explain the high variety of Cl levels determined at ZT2 and ZT12?
4. In Figure 1 approximately 10 times more neurons were recorded for ZT17 compared to ZT5, while the number of mice imaged is not even twice as many. How can this difference be explained.
5. Figure 1A the color code does not indicate what it presents and the scale bar 50 μm is called calibration bar in the figure legend and states to be 40 μm .
6. Figure 1C shows the effect of bumetanide, please also show an example for ZT5 and indicate how many of the cells responded to the drug treatment. The methods explain the application of bumetanide in many different ways. An illustration for each experiment would be helpful.
7. The data presented in Figure 2 needs some statistical analysis and comparison of ZT5 and ZT17. A similar analysis compared to Figure 3 would be helpful to compare the data that was obtained.
8. Figure 4 does not show example traces for ZT5 with bumetanide, and it is not clear which anesthetic was used in these recordings.
9. In Figure 4 the bumetanide effect on ZT5 shown in (e) seems to increase the peak power in half of the mice, while it is decreased in the other half. Please indicate in (d) which data points belong to the same animal.
10. Figure 5 (c) shows an illustration of the experimental setting indicating that the mice are awake. Please correct this to a closed eye, anesthetized mouse.
11. The data in Figure 5 and 6 seems to be that same, the data could be combined or shown in supplemental figures.
12. Statistic analysis of the data presented in Supplementary figure S4 is needed.

Reviewer #3:

Remarks to the Author:

'Diurnal rhythm in cortical chloride homeostasis underpins functional 1 changes in visual cortex excitability' by Pracucci et al., revealed a diurnal pattern of intracellular Cl⁻ of cortical pyramidal neurons by using ClopHensor in vivo imaging. They then found that the altered surface level of KCC2 and phosphorylation level of NKCC1 at different time of the day correlate with the fluctuated [Cl⁻]_i. They further showed that the altered [Cl⁻]_i correlate with the visual evoked gamma power and the

susceptibility to 4-AP-induced seizure activity, both of which can be affected by applying bumetanide to block NKCC1. Finding of the diurnal changes in intracellular Cl⁻ is interesting and coincides with previous study by Alfonsa et al.. The functional implication of such Cl⁻ changes provides explanation for the daily changes in cortical function and pathology. However, some major questions need to be addressed before being further considered for publication.

1. ClopHensor was expressed by two means, neither of which is selective for pyramidal neurons. However, authors claimed that they were imaging pyramidal neurons only, which will require extra proof such as immunostaining. Are other cell types such as inhibitory neurons also changing intracellular Cl⁻ during the day?
2. The author showed imaging results of different mice instead of monitoring the same population of neurons at different time points under anesthesia. The altered [Cl⁻]_i could be complicated by varied basal level of Cl⁻. Anesthetization has also been well known to affect Cl⁻ concentration by targeting Cl⁻ channel such as GABAAR. Monitoring of Cl⁻ signal of the same population of neurons across different time point in fully awake mice will provide necessary evidence to strengthen the finding.
3. Figure2 shows the removal of membrane KCC2 while change in phosphorylation of total NKCC1 as a mechanism explaining the Cl⁻ alteration. Is there any change in the membrane fraction of NKCC1, which is supposed to be the functional one? What may drive the internalization of KCC2 at ZT17 and NKCC1 phosphorylation is regulated? Are these phenomena sleep or circadian-dependent? The authors should at least provide some mechanistic study.
4. The physiological relevance of Cl⁻ with visual evoked gamma and seizure susceptibility may also be explained by altered synaptic inhibition at different time of the day. The bumetanide may target inhibitory neurons by elevating their excitability instead of pyramidal [Cl⁻]_i. Therefore, their claim of pyramidal Cl⁻ contributes to the gamma and seizure requires further evidence.
5. Several mismatches between figure1 index and the result section.

We would like to thank the reviewers for their time and efforts to critique our work and for their truly helpful comments. We have made extensive changes to the manuscript, adding new data, editing text in multiple places, and adapting multiple figures (new versions of all the figures, except figure 2 - for which we have created a new explanatory figure in supplementary information - and we have also added multiple new supplementary figures). We have submitted a tracked version of the edited manuscript and below, we describe what changes have been made in response to each of the specific points made by the reviewers. For clarity, we have italicized the reviewers' comments and keep these in full, and we have interspersed our responses (non-italicized).

REVIEWER COMMENTS

Reviewer #1 (Remarks to the Author):

I would like to congratulate the authors on this excellent contribution, that I believe will become a citation classic, since it dramatically extends understanding of how neuronal chloride homeostasis, and its temporal variation, impacts upon neuronal electrophysiology in the adult mammalian brain. Using a very sophisticated fluorescent probe for neuronal chloride levels, previously developed by Ratto, they show daily variation in pyramidal neurons in living mice being higher during the night/active phase. This extends previous findings (from SCN neurons by Jen Evan's and Doug McMahon's labs, and assorted non-neuronal cells by Stangherlin/O'Neill labs) into the cortical neurons of adult brain. In itself, this is a major findings, since daily variation in the chloride reversal potential would be expected to lead to daily variation in the potency and direction of GABAergic neurotransmission. They go on to demonstrate that variation in Chloride levels is consistent with differential daily variation in NKCC/KCC activity.

The authors then test two functional predictions that the time of day-dependent variation in 1) gamma-band synchronisation by visual stimulation in visual cortex and 2) susceptibility to epilepsy can be selectively blocked by bumetanide at night (inhibits NKCC mediated chloride efflux). Overall this is excellent hypothesis driven neuroscience that can be expected to have a broad impact, because it raises as many follow-up questions. I have a few comments, as follows:

Major points

1. Although it is technically correct that these experiments were performed under diurnal conditions, many (potentially) interested readers will not know the word or may be confused by knowledge that the normal behaviour of a mouse is nocturnal. To more clearly communicate their findings to a wide audience, I would strongly encourage the authors to use "daily" in place of "diurnal", both in the title and throughout, since this is unambiguous and more accessible.

We have changed all the instances of diurnal, including in the title, as advised.

2. There is no clear justification for the animal numbers used in each experiment or the number of neurons recorded from each mouse. Neither is there any clear explanation of whether there was consistently 50:50 male:female mice, and if not why not. This is not a fatal problem but must be addressed in the methods section.

We had shown the sex of the imaged mice in the supplementary data, and this was almost exactly 50:50 (15 F, 14 M) but we accept that this information was missing elsewhere, and we now provide more information. In short, in none of the metrics did we see any significant effect of sex of the animal (the sensitivity to 4AP showed a trend towards earlier seizures in females). Importantly, with respect to the main results we present regarding the day/night difference, both male and female animals show the same changes.

3. The text and data presentation explaining Fig 5/6 need to be improved. The text states that two different experimental paradigms were employed: injection of 4-AP into neocortex vs

injection of 4-AP into neocortex followed by direct application of bumetanide. There does not appear to be a "vehicle" control for the latter. This is a concern because, in principle, that the act of applying the same solution without bumetanide might also affect seizure susceptibility during the night. I really would expect this control to be performed in order to have confidence in the effect of bumetanide.

The main solvent for bumetanide is saline, and saline was applied to the surface of the brain in every animal, once the craniotomy is performed, to keep the brain moist. This procedure was followed identically for all experiments including those including the bumetanide application, so that is a form of vehicle control. There is, however, a small additional supplement of DMSO (~0.08%), since bumetanide is dissolved in DMSO for the stock solution. We had controlled for this trace amount of DMSO in our previous work (Sulis Sato et al, PMID 28973889), and found no effect on the chloride level in neurons. We have now performed the same control in the night-time experimental group in seizure induction assay. These DMSO-treated mice showed the same rapidly evolving epileptic activity as the control group (no bumetanide), in stark contrast to the almost complete absence of epileptic activity in the bumetanide + DMSO experiments, indicating that the altered course of evolving epileptiform activity was indeed due to the bumetanide and not the solvent. This data is shown in a new version of figure 5, and this new detail about the vehicle control is referenced in the figure legend.

Minor points:

Interaction with anaesthesia?

The reviewer is likely to be correct that some anaesthetic regimes may affect pH and chloride, and of course they affect neuronal activity (otherwise they would not be anaesthetics) – however, the relationship is complex, and is a topic of on-going investigation. For the current study, however, in all facets of the study, we compared the day / night measures in identical preparations, including the choice of anaesthetic agent. Very slightly different anaesthetic regimes were used at the two centres, but notably, the data at both sites coincided – we have added some text to this effect (page 5, lines 159 and following). Furthermore, note that day / night differences in inhibitory function are also seen in the awake animals (Figure 3) and in cortical brain slices (Alfonsa et al, PMID 36510112), both corroborating the findings in anaesthetized animals, providing strong support that the day/night modulation is a robust phenomenon. Finally, the starting point of our collaboration in this study was to corroborate the Pisa imaging findings by replicating these in Newcastle. These used a slightly different mouse strain, a different means of introducing the biosensor, and also a slightly modified anaesthetic regime, and yet the Cl⁻ imaging data at both sites, at ZT5 and ZT17, were exactly in line with each other. This is actually a very important validation that is rarely done within a single study.

It would be helpful to have a cartoon schematic to explain the experimental setup and data processing that allows one to get to Fig3c. I know this is partly explained in the text and partly in the methods section, but it would help the reader enormously to see the workflow diagrammatically.

We provide a schematic (Supplementary Figure S8).

It was not immediately clear to me how the analysis presented in Fig 4b/c differs from that in Fig3c. Please can this be explained more clearly of the method of analysis harmonised?

In short, Figure 3 shows data from non-anaesthetized (awake) animals and were paired data sets (day and night data) in which the same mouse was recorded at the two different time points. We could not apply bumetanide to the cortical surface of these mice because the implant of the chronic electrodes did not allow a direct access to the cortex. Since it has

already been described by Veit (2017) that the grating-induced oscillations are resistant to anaesthesia, we opted for the direct application of bumetanide during these recording with the same modalities used in the imaging experiments. The recordings from the two sets of experiments have been performed with electrodes of different size and impedance resulting in different signal to noise ratio. For this reason, we opted to perform a slightly different analysis. We have added text in the Methods section explaining this (section on "Electrophysiology analysis").

Line 40 - adult mouse neocortex not mice

Changed

Line 54 - "Basic animal research has further documented differences in the level of neuronal..." ... presumably means "Basic animal research has further documented time -of-day differences in the level of neuronal..."

Changed

Line 95/129 and elsewhere - "the 24hr period" should be "day/night cycle"?

Changed

Line 105 - worth justifying choice of neuron/location?

Done

Line 115 - daily rest/activity cycle instead of diurnal cycle

Changed

Line 131 - mouse's not mice's

Changed to "murine"

Line 484 - mouse pups

Changed

Line 209 - means Fig 2c?

Figure 3c (not 2c and not 3b as was in the previous submission) - changed

Line 237-9 - sentence ends with a reference to Fig5b, but isn't clear how the data shown in 5b relates to preceding text since 10 min baseline is not shown, neither are the characteristic UP states that are referred to. Please clarify.

We have included examples of the UP states for all three conditions. We also show examples epochs from all recordings in a new supplementary figure, at lower resolution (Supplementary Figure S11).

Line 285 - NKCC and KCC are meant to be electroneutral transporters so from the context I presume the authors mean "accompanied by daily redistribution of cation counterions" rather than the rather vague "affected by circadian redistribution of other ionic species" ...if the same type of process described by Stangherlin is operating here?

Changed

Line 302 - should also refer to prior work on circadian rhythms in SCN neuronal chloride levels by Jen Evans and Doug McMahon's lab.

We have included these references now.

Discussion - a sentence or two about what might generate these daily changes in Cl levels would not go amiss.

In other systems, the mTOR pathway has been implicated in daily modulation of ionic levels (Stangherlin et al, 2021), and it will be interesting to clarify how that relates to neuronal activity patterns and imposed disruption of circadian rhythms. We added some text (lines 300-304).

Fig 2a - SDS-PAGE was run under reducing conditions, how are these dimers occurring/being detected?

Protein samples were prepared in NuPAGE™ LDS Sample Buffer (reducing conditions), and were boiled in the same sample buffer at 75 °C for 10 minutes, and SDS-PAGE was indeed run under reducing conditions, which should, in theory, dissipate dimers. In practice, however, we find that it is still possible to detect trace levels of KCC2 dimers, as evident in the images, and also in previous studies (e.g. PMIDs 26126716, 29176664, 31911626, 31615899, 31615901).

Fig 2b - might be clearer to use pm-KCC2 and all-KCC2 or similar, and do the same for NKCC1 for consistency. Either way, which was used for the denominator in the ratios in top right and middle left.

We followed a methodology that we developed in 2015 (PMID: 26126716), and which has been used in multiple publications since, both by ourselves and other research groups (see refs above). Having said that, the reviewer's suggestion has merit, in that it would also indicate the relative activity of the two co-transporters, and perhaps be a more intuitive presentation for the reader; we will bear this in mind in future studies. Following the reviewer's second suggestion, we made a new supplementary figure (Supplementary Figure S7) to indicate the numerator and denominator in each analysis in Figure 2b.

Fig 3a/b & 4a. Heat maps are missing a y-axis label

We have added the y-axis labels (Frequency (Hz))

Fig 4b-e. Please report number of animals

The missing information was added in the figure legend.

Fig 6. Please show all traces in a supplementary figure, not just those with an ictal event.

We have changed this figure (which is now Supplementary figure S11) to show the pre-4AP recordings and the final 5 min epoch at the end of the recorded period (55-60mins), as well as higher resolution views of the first seizures for all the relevant cases.

Reviewer #2 (Remarks to the Author):

Nature Communications manuscript NCOMMS-22-46047-T

In the manuscript Diurnal rhythm in cortical chloride homeostasis underpins functional changes in visual cortex excitability the authors present interesting new data showing that Cl in cortical neurons differs significantly at different times of the day. The time dependent Cl concentration is correlated with whole brain phosphorylation levels of NKCC1 and membrane expression of KCC2. Furthermore, application of the NKCC1 targeting drug bumetanide affects visual processing and induced epileptic activity. While the basic finding of this study is highly relevant

and new, the underlying mechanism which is presented is questionable based on the data presented.

Major comments:

1. Western blot analysis in Figure 2 was performed on whole brain tissue without discussing which cell types might contribute to this result. While there is a difference in membrane expression of KCC2, NKCC1 only differs in phosphorylation levels. Please discuss in more detail how different phosphorylation can affect the activity of NKCC1. The amount of membrane bound NKCC1 has not been determined.

The reviewer is correct that the western blot data merges all the cell classes, but it is also the case that the pyramidal population represents around 80% of neurons in rodent cortical networks. We accept that this is a loose association, but it is the best that we could achieve at this stage and is not the critical element of the study. We were, however, careful not to overstate our results, for exactly the reasons raised by the reviewer. Regarding the membrane bound NKCC1, we chose not to do this because in our previous studies, we have found that changes in NKCC1 activity, seen in various pathological states, tend to reflect changes in its phosphorylation state, rather than the membrane bound NKCC1 levels (Zhang et al, 2016 *Sci.Rep*, **6**, 35986; Bhuiyan et al, 2022, *Stroke* **53**, 1720; Karimy et al, 2017, *Nature Med*, **23**, 997). Unfortunately, JZ has relocated to China since this part of the study was done, and it is not something we can do easily at this time. We have made small edits to the text to explain why we did these assays, and then also in the discussion to flag up the difference between the cell-class specific imaging data and all the other assays (also the physiological assays) which do not have the same single cell, or cell-class, resolution.

2. The authors aim to manipulate neuronal Cl using bumetanide. This needs to be shown by imaging neuronal Cl with and without bumetanide. In the following experiments bumetanide was applied for different amounts of time, please show for the different duration (40 and 60 min) how neuronal Cl is affected by bumetanide.

Figure 1 had the bumetanide figure for the ZT17 state. We had previously omitted to show the ZT5 equivalent data, because bumetanide did not change anything, but for completeness, we have now added this panel.

The experiments of figures 1, 4 and 5 were all started between 40 and 60 min after the treatment beginning. We have added a new supplementary figure (Fig S6), showing repeated Cl_oHensor imaging data at different time intervals following bumetanide treatment that show that chloride levels appear to reach a new steady state approximately 40 minutes after bumetanide application.

3. The major part of the presented data was performed under two different kind of anesthesia, which are both known to affect anion channel in different ways. It is crucial to highlight which anesthetic was used in which experiment, compare neuronal Cl- under both anesthetics and discuss how those might affect the presented data.

The Pisa and Newcastle imaging and 4-AP experiments were performed using very slightly different protocols (see our earlier comment to a comment of reviewer 1), including the anaesthetic regimes. In all cases we have made efforts to be clear about which anaesthetic was used. It was notable therefore that there was no difference between the data acquired at the two sites, and both showed very large modulation between day and night.

Minor comments:

1. Two different mouse lines were used. Which line was used for which experiment, are Cl dynamics and expression pattern for NKCC1 and KCC2 the same?

Chloride imaging was performed on both CD-1 IGS mice and C57BL/6 mice, and the electrophysiology and western blot studies were performed on C57BL/6 mice only. This is

stated at the start of the Methods section. Importantly, there was no difference in the Cl⁻ measures between the two strains.

2. Expression of the Cl sensor ClpHensor was obtained by two different ways, in utero electroporation and virus injection. IHC staining showing expression pattern and cell type specificity are needed.

We provide this in a new figure in the supplementary information (Supplementary Figure S1) for the viral injections where the identity of the transfected neurons was confirmed by the pyramidal cell marker CamKII. Note that the specificity of labelling pyramidal cells using tightly timed in utero electroporation was established a while ago (Hatanake et al, 2004 – this reference has been added to the manuscript) and has been used in multiple papers since. It is important to note that the data from both approaches were entirely consistent, which is a key control for the two approaches for introducing the transgene.

3. The activity pattern of the mice used for Cl imaging was tracked, does activity correlate with the Cl level determined? Can different activity levels explain the high variety of Cl levels determined at ZT2 and ZT12?

The activity patterns were tracked for periods prior to the experimental recordings, simply to check that each individual mice did have an appropriate night / day behaviour. We did not, however, do the imaging during the periods of behaviour; rather, in all cases, the mice were anaesthetized. This should normalise the neuronal activity, so we do not think that the level of neuronal activity at the time of the imaging underlies this difference. Interestingly, just last month, Alfonso et al published that a qualitative similar daily modulation is also apparent in brain slices prepared at different times of day. Of course, in brain slices, activity is extremely reduced relative to the in vivo state. In that paper, they suggested that activity levels might be a factor, but the activity dependent changes in Cl⁻ that were reported initially by Thompson and Gahwiler, and then by multiple other groups since including us (see figure 8 of Sulis Sato *et al.* PNAS 2017), is corrected quite soon after activity is curtailed, and so we do not fully agree with the interpretation presented by Alfonso et al.

4. In Figure 1 approximately 10 times more neurons were recorded for ZT17 compared to ZT5, while the number of mice imaged is not even twice as many. How can this difference be explained.

This is not the case - the reviewer has misread the numbers in the legend; we sampled 663 neurons from 9 mice at ZT5 and 1051 neurons from 8 mice at ZT17 (less than a 2-fold difference). We had lower sample numbers at ZT2 and ZT12, but these represented transition time points, and the main interest here was in the ZT5 and 17 datasets.

5. Figure 1A the color code does not indicate what it presents and the scale bar 50 μm is called calibration bar in the figure legend and states to be 40 μm.

Thank you for pointing out the discrepancy. We have checked the scale bar, and it is indeed 40 μm, and have corrected this.

6. Figure 1C shows the effect of bumetanide, please also show an example for ZT5 and indicate how many of the cells responded to the drug treatment. The methods explain the application of bumetanide in many different ways. An illustration for each experiment would be helpful.

We have added the requested panel (ZT5). We have not indicated the number of responsive cells since, as we do not have repeated measurement before and after bumetanide we have no data-based method to provide a proper estimate at the level of single cell. For this reason, we opted to study the population behaviour.

Bumetanide was actually applied in the same way in all cases, but our descriptions of this were inconsistent. We have corrected this issue, in the methods section.

7. *The data presented in Figure 2 needs some statistical analysis and comparison of ZT5 and ZT17. A similar analysis compared to Figure 3 would be helpful to compare the data that was obtained.*

The data in Figure 2 is analysed using Wilcoxon rank sum test, and we have added an explanatory supplementary figure to explain how the ratios in each panel were derived.

8. *Figure 4 does not show example traces for ZT5 with bumetanide, and it is not clear which anesthetic was used in these recordings.*

We have added a panel to show a day-time bumetanide recording. Detail about the anaesthetic is included in the methods section.

9. *In Figure 4 the bumetanide effect on ZT5 shown in (e) seems to increase the peak power in half of the mice, while it is decreased in the other half. Please indicate in (d) which data points belong to the same animal.*

We have added vectors linking the paired points, in both day and night data sets. This is shown in Supplementary Figure S9.

10. *Figure 5 (c) shows an illustration of the experimental setting indicating that the mice are awake. Please correct this to a closed eye, anesthetized mouse.*

This is a cartoon and does not imply the animal is awake – in any case, mice usually have their eyes open under anaesthesia, and in fact they exhibit responses to the presentation of patterned visual stimuli. We hope that this is not problematic and so we have not changed the image.

11. *The data in Figure 5 and 6 seems to be that same, the data could be combined or shown in supplemental figures.*

They are indeed the same. They are shown in separate figures simply because we wanted to show the recordings in sufficient detail such that readers could appreciate that the ZT17 seizures were typically more intense events. If they were incorporated into Figure 5, the panel would be too small to appreciate this. Another reviewer asked for additional data to be shown in figure 6, which we have added. We have however, now moved this to be supplementary figure S11.

12. *Statistic analysis of the data presented in Supplementary figure S4 is needed.*

Done

Reviewer #3 (Remarks to the Author):

'Diurnal rhythm in cortical chloride homeostasis underpins functional 1 changes in visual cortex excitability' by Pracucci et al., revealed a diurnal pattern of intracellular Cl⁻ of cortical pyramidal neurons by using Clophensor in vivo imaging. They then found that the altered surface level of KCC2 and phosphorylation level of NKCC1 at different time of the day correlate with the fluctuated [Cl⁻]_i. They further showed that the altered [Cl⁻]_i correlate with the visual evoked gamma power and the susceptibility to 4-AP-induced seizure activity, both of which can be affected by applying bumetanide to block NKCC1. Finding of the diurnal changes in intracellular Cl⁻ is interesting and coincides with previous study by Alfonsa et al.. The functional implication of such Cl⁻ changes provides explanation for the daily changes in cortical function and pathology.

However, some major questions need to be addressed before being further considered for publication.

1. *ClpHensor* was expressed by two means, neither of which is selective for pyramidal neurons. However, authors claimed that they were imaging pyramidal neurons only, which will require extra proof such as immunostaining.

We appreciate that we should have provided more detailed background information to make this point clear. The *in utero* electroporation technique was first demonstrated to label pyramidal cells specifically in 2004 (Hatanaka et al, reference added to the paper), and has been replicated in multiple studies (including several of ours – PMIDs 22805567, 26844428). We have added the Hatanaka reference with a statement point out the identity of the transfected neurons. The second approach used an *Emx1-cre* line that does label some glia early in life, but by adulthood, the labelling is specific to pyramidal cells, as described in the original description of this promoter (Gurski et al, 2002). We have provided immunohistochemistry images to support this (Supplementary figure 1) and we have clarified the point in the main text.

Are other cell types such as inhibitory neurons also changing intracellular Cl⁻ during the day?

Yes, we believe so, but this is work in progress, which raises additional mechanistic issues that would require further work to clarify. We are disinclined to include it in this study, in part because it would delay publication of this first paper, and secondly because it is very important for the career progression of the postdocs that they have papers where they take the main credit – we already have 4 first authors listed here, and it is important for their careers that we keep some separation of studies, instead of merging all the work together.

2. *The author showed imaging results of different mice instead of monitoring the same population of neurons at different time points under anesthesia. The altered [Cl⁻]_i could be complicated by varied basal level of Cl⁻. Anesthetization has also been well known to affect Cl⁻ concentration by targeting Cl⁻ channel such as GABAAR. Monitoring of Cl⁻ signal of the same population of neurons across different time point in fully awake mice will provide necessary evidence to strengthen the finding.*

This is an important question, but we have not managed to achieve chronic repeated imaging of the same neurons at very different times. These experiments are of course very important and to that effect we are developing a new generation of Cl sensors that have better sensitivity and lower pH dependency and that are ideally designed for chronic imaging through a wearable miniscope. However, these experiments are still in the future and are not currently possible with the present technique.

3. *Figure 2 shows the removal of membrane KCC2 while change in phosphorylation of total NKCC1 as a mechanism explaining the Cl⁻ alteration. Is there any change in the membrane fraction of NKCC1, which is supposed to be the functional one? What may drive the internalization of KCC2 at ZT17 and NKCC1 phosphorylation is regulated? Are these phenomena sleep or circadian-dependent? The authors should at least provide some mechanistic study.*

See also our response to the similar comments from Reviewer #2. We were following protocols that have been used on multiple occasions both by the biochemist leading this element of the study (Zhang et al, 2016 *Sci.Rep*, 6, 35986; Bhuiyan et al, 2022, *Stroke* 53, 1720; Karimy et al, 2017, *Nature Med*, 23, 997). Zhang has recently moved to China and we could not perform further studies on this within the time-frame for the turnaround of the revisions. We are currently working towards dissecting out the relative influence of sleep / intrinsic circadian factors.

4. *The physiological relevance of Cl⁻ with visual evoked gamma and seizure susceptibility may also be explained by altered synaptic inhibition at different time of the day. The bumetanide may target inhibitory neurons by elevating their excitability instead of pyramidal [Cl⁻]_i. Therefore, their claim of pyramidal Cl⁻ contributes to the gamma and seizure requires further evidence.*

We agree that the causal chain of events is not entirely proven, but the point of our paper is that from day to night, there are parallel changes in multiple assays, all of which show the same pattern of sensitivity to bumetanide, namely an absence of effect at around ZT5 and a large effect at around ZT17. Another paper that has been published while this paper has been in review (Alfonsa *et al.*) report similar findings, using very different methodologies, adding confidence. It is quite difficult to explain the effect of bumetanide upon the directly visualised changes in [Cl], in pyramidal cells (Figure 1), whereas the explanation we proffer actually covers all the data set. The fact is that one could also conceive other causal explanations too. We have adapted the discussion to try to avoid being too dogmatic in our interpretation of the data, and indeed we finish by listing other relevant papers that present findings that will also interact with the changes in GABAergic function that we report.

5. Several mismatches between figure1 index and the result section.
We have corrected all the figure references

Reviewers' Comments:

Reviewer #1:

Remarks to the Author:

I am content that the authors have satisfactorily addressed all the major reviewer comments and further improved the clarity of the manuscript. I congratulate them on this excellent piece of work.

Reviewer #2:

Remarks to the Author:

Nature Communications manuscript NCOMMS-22-46047-T

The authors added a noticeable amount of new data to the manuscript which improved the manuscript. However, not all concerns have been addressed sufficiently.

1. While in the results section explain the reason for choosing different ways to quantify expression levels of NKCC1 versus KCC2, the discussion is still lacking a more detailed interpretation and reasoning for how NKCC1 phosphorylation levels and KCC2 membrane localization account for the observed differences in chloride levels. This information seems to be crucial for the study, as the effect of bumetanide is based on these findings. Could the authors explain their model better?
2. Figure S6 shows the bumetanide effect over time which reaches a new steady state after 50-60 min. However, in Figure 1 bumetanide was incubated for 40 min. Thus, the effect on chloride is therefore likely underestimated. Or is there a different reason for different incubation times? This is a concern and Figure 5 is missing information about the incubation time.
3. Figure S4 compares the effect of the different anesthetics used. However, the discussion does not address the fact that anesthetics in general have massive effects on brain physiology and might affect neuronal chloride.
4. In the rebuttal letter the authors provide interesting arguments about the activity independence of neuronal chloride data presented. With regards to the recently published paper by Alfonsa et al, which claims activity pattern rather than time of day is the key factor determining neuronal chloride, it is crucial that the authors add their arguments to the discussion.
5. The authors claim that they do not have repeated measurements before and after bumetanide application. However, Figure 1B shows exactly that. If the data in Figure 1C is not obtained as indicated in Figure 1B more detailed information is needed to avoid misleading the reader. The figure legend F1C needs to be corrected: Bumetanide: chloride decreased in all neurons in the field at ZT17.
6. Even though it seems like all information about anesthetics used is provided somewhere in the method section, it is hard to follow which methods were used to obtain which data set. In this manuscript, where different anesthetics have been used with different protocols in different labs it is crucial to be highly transparent and clear about the methods. All information should be provided with the data and not be hidden in the method section. The same is true for bumetanide applications and incubation times.

Reviewer #3:

Remarks to the Author:

The author provided sufficient explanation for most of my technical concern. It will be great if they can address my 3rd question more in detail. Some of the experiments can be easily carried out. For example, the sleep and circadian dependency can be simply teased out by doing an acute sleep deprivation. The functional correlation with daily intracellular Cl⁻ change can also be supported by more solid and complete experiment such as adding bumetanide at zt5 as well, and using KCC2 blocker to convert ZT5 to ZT17.

We thank the reviewers for taking the time and trouble to critique our manuscript, and for their constructive comments. In response to these, we have performed several new experiments, including to examine the effects on intracellular chloride of extended dark housing and of KCC2 blockade. These experiments took longer to do than we had initially expected, but the results are very interesting and definitely help clarify important points of the underlying biology. We have split Figure 1 into two different parts for clarity and we have added several new panels, so the MS now includes 6 main figures and 11 supplementary figures. The abstract and introduction have been modified to reflect the new data.

We indicate what has changed from our previous submission, in our response to the individual comments provided by the Reviewers, below. We include their original critiques (italicized and in blue) and intersperse our responses between.

We provide a clean version of the new manuscript, and one with all changes tracked. Note that the line numbers refer to the clean version, and not the tracked version (where they are obvious anyway from the tracking).

Reviewer 1 had no further comments.

Reviewer 2

The authors added a noticeable amount of new data to the manuscript which improved the manuscript. However, not all concerns have been addressed sufficiently.

1. While in the results section explain the reason for choosing different ways to quantify expression levels of NKCC1 versus KCC2, the discussion is still lacking a more detailed interpretation and reasoning for how NKCC1 phosphorylation levels and KCC2 membrane localization account for the observed differences in chloride levels. This information seems to be crucial for the study, as the effect of bumetanide is based on these findings. Could the authors explain their model better?

We have performed new experiments using VU0463271 at ZT5, showing changes in both the directly measured $[Cl^-]_i$ and the downstream physiological effects. These data help explain the complementary effects of the two co-transporters on intracellular chloride and on the downstream regulation of excitability. We have added a schematic (Figure 2f) to provide a summary of the imaging results and to introduce the biochemistry experiments. We have also added text in both the results and the discussion relevant to the new data we present. The summary of these changes is that, relative to each other, NKCC1 activity is high during the day and KCC2 activity is high at night.

2. Figure S6 shows the bumetanide effect over time which reaches a new steady state after 50-60 min. However, in Figure 1 bumetanide was incubated for 40 min. Thus, the effect on chloride is therefore likely underestimated. Or is there a different reason for different incubation times?

We based our assessment of the steady state effect of bumetanide upon the median values (supplementary figure 6), although the Reviewer correctly observes that there is still some excess variance in the 40-50 min measurements. Given, though, that the median is the best

estimator of the population (see our previous publication, Maset et al, 2021, PNAS), we chose to use that as the cut-off time.

Regarding the precise timing, it is helpful to understand that due to the nature of these *in vivo* experiments, in which imaging at multiple wavelengths of each field takes several minutes, it is impossible to provide an accurate timing of the data collection. So we simply stated that the minimal incubation time is 40 mins. In the revised MS we clarified this point at lines 170f.

Again, the Reviewer is correct to observe that including data points made between 40-50 minutes post-bumetanide treatment could yield slightly weaker effect, this issue is minimized by using median values, and this is also offset by increasing the sample size. Inclusion of weaker pharmacological effects would only have been an issue if we were reporting null results, but in fact, in all cases, we were reporting highly significant differences. We felt reassured therefore that the changes that we report are all robust.

Figure 5 is missing information about the incubation time.

The incubation time of the co-transporter inhibitors was provided in the text (lines 294f; note that this is now Figure 6) but the Reviewer is correct that it is helpful to include this also in the figure legend. We have done so now as shown at lines 524.

3. Figure S4 compares the effect of the different anesthetics used. However, the discussion does not address the fact that anesthetics in general have massive effects on brain physiology and might affect neuronal chloride.

The Reviewer is correct to say that anaesthesia may acutely affect the exact levels of Cl, and this will be an interesting future topic of investigation, when we established how to perform quantitative Cl imaging in behaving mice. However, for the current study, the key point of Figure S4 is that our finding about Cl modulation was replicated in two laboratories, despite some minor differences in anaesthetic regimes, dictated by the preferred anaesthetic regimes in Italy and the UK. We are rather proud that we were able to replicate this important result, under subtly different protocols, which reassured us, and we hope the readers too, about its robustness. Notably, our conclusions about the biology have since been further reinforced by the publication from Akerman's group, in Oxford, using other methodology complementary to ours. Thus, equivalent results regarding the modulation of chloride levels have been found now in 3 labs – in our own, in Pisa and Newcastle and Akerman's lab in Oxford – to be published in two complementary papers, that furthermore, both come to the same broad conclusions. Importantly, each laboratory followed internally consistent protocols in day and night.

We add a comment to this effect in the discussion, as requested at lines 322f. We note that the reviewer asked about anaesthesia in the first iteration of reviews, and we apologize for having slightly deflected the question.

4. In the rebuttal letter the authors provide interesting arguments about the activity independence of neuronal chloride data presented. With regards to the recently published paper by Alfonso et al, which claims activity pattern rather than time of day is the key factor determining neuronal chloride, it is crucial that the authors add their arguments to the discussion.

The question of whether the observed Cl change arises from circadian modulation, or secondary to modulation of activity, or a combination of both, is indeed an important and

interesting one. To address this, we performed a new set of experiments in which we kept a cohort of mice under extended darkness, while monitoring their behavioural activity, and then imaged intracellular Cl⁻ (see also our response to point 1 of Reviewer 3). The new data show that despite the animals being more active during this period of extended darkness, the Cl⁻ modulation appears unchanged from what is seen in animals in a normal light cycle. This strongly suggests that the baseline changes we observed are due to an autonomous circadian regulation, rather than to acute changes due to activity. Incidentally, this interpretation is consistent with what has been observed in basically all cell types from yeast to cardiomyocytes as shown in Stangherlin et al (Nature Comm 2021). It should be noted that the manipulation performed on mice in the Alfonsa paper (where they provided continual experimenter-driven stimulation of the mice during the period of extended darkness – termed “sleep deprivation”) is more invasive and potentially more stressful than simply leaving the light off. Therefore, a direct comparison between our results is not possible. This raises multiple other questions, as all interesting science does, but a full dissection of these issues lies beyond the scope of this study. We have described the new data in the revised figure 1 and at lines 159f and 331f.

5. The authors claim that they do not have repeated measurements before and after bumetanide application. However, Figure 1B shows exactly that. If the data in Figure 1C is not obtained as indicated in Figure 1B more detailed information is needed to avoid misleading the reader. The figure legend F1C needs to be corrected: Bumetanide: chloride decreased in all neurons in the field at ZT17.

We apologize because one of our responses to the previous set of reviews was somewhat ambiguous, and this is where the confusion arose. We did indeed perform paired measurements before and after bumetanide treatment both at ZT5 and ZT17. Indeed, the data presented in Figure 2b can only be obtained performing imaging of the same neurons before and after bumetanide treatment. We have corrected the figure Figure 2b legend accordingly.

6. Even though it seems like all information about anesthetics used is provided somewhere in the method section, it is hard to follow which methods were used to obtain which data set. In this manuscript, where different anesthetics have been used with different protocols in different labs it is crucial to be highly transparent and clear about the methods. All information should be provided with the data and not be hidden in the method section. The same is true for bumetanide applications and incubation times.

We have added this information, as requested in all pertinent points in the main text.

Reviewer #3 (Remarks to the Author):

The author provided sufficient explanation for most of my technical concern. It will be great if they can address my 3rd question more in detail. Some of the experiments can be easily carried out. For example, the sleep and circadian dependency can be simply teased out by doing an acute sleep deprivation. The functional correlation with daily intracellular Cl⁻ change can also be supported by more solid and complete experiment such as adding bumetanide at zt5 as well, and using KCC2 blocker to convert ZT5 to ZT17.

We are grateful for the Reviewer's suggestions as we believe that they add greatly to the study. The new data are now present in Figure 1, 2 and 6 they are described in the

text and also in a modified Abstract. We have addressed both points raised by the reviewer by answering two questions:

1) is the Cl decline between ZT17 to ZT5 due to light onset and the associated change in behavioural activity, or does it occur independently of the light onset?

This was addressed by rearing mice in a 7:00 – 19:00 light regime for several days in a quiet environment before switching to continuous darkness for 4-5 hours starting at ZT0 before imaging at ZT5. During this rearing, the mouse was video-recorded, and locomotor activity was monitored by on-line analysis of the video stream. In these new experiments (6 mice, 588 imaged neurons) we determined that 4-5 hours of dark rearing prolonged activity in all mice and reduced immobility in 4 out of 5 mice. Remarkably, despite the persist behavioural activity of the mice, the $[Cl^-]_i$ levels followed that of the control mice, that were kept in the normal light cycle (ie with light turned on at ZT0). This data shows that the Cl cycle is independent of light exposure at ZT0, and the increased locomotor activity caused by prolonged darkness. These new data have been presented in the new Fig 1 and are described at lines 159f and 331f.

2) Does the inhibition of KCC2 at ZT5 convert cortical excitability to the level present at ZT17?

As requested by the Reviewer, we did these exact experiments, testing the effects of the KCC2 blocker, VU0463271, at ZT5, first, by imaging $[Cl^-]_i$ (Figure 2d-e; text at lines 185-191), and second using 4-AP (Figure 6; text at lines 313f). VU0463271 caused $[Cl^-]_i$ to rise by about 8 mM, while in the latter experiments, it converted the low propensity to seizure observed at ZT5 to match that seen at ZT17, shortening the latency to the first seizure and dramatically increasing the cumulative pathological electrophysiological discharges. So, for both experiments, the effect of blocking KCC2 activity in ZT5 mice appears to convert them to the ZT17 state.

Reviewers' Comments:

Reviewer #2:

Remarks to the Author:

The authors have done a good job responding to the critique. Only remaining comment is that the labeling of the scheme shown in figure 2f could be better.

Reviewer #3:

Remarks to the Author:

Thank you for addressing my previous concerns. With the additional evidences provided by the authors, this manuscript is much strongly presented. Couple things that need further verification: 1) line 177-178, why are different statistics used for ZT5 and ZT17 experiments? Are they both paired comparison? 2) line 182, Signed rank test is designed for paired test, but the ZT17 pre and post bumetanide cell number are different. 3) The prolonged dark exposure does answer that the reduction in intracellular Cl⁻ is not triggered by light. But since this is not a complete sleep deprivation, some amount of sleep was still preserved though they did show activity level was increased in these DE mice. Therefore, they still can't conclude that the change in Cl⁻ is mainly time-dependent. They should at least discuss the potential interplay between circadian and sleep on such regulation.

Otherwise, the story reads good to me.

ANSWERS TO REVIEWERS' COMMENTS

We are grateful to the Reviewer for the help they provided with their feedback that improved markedly the scope and strength of our work. In the following we respond to all their final comments reported verbatim in italics. We also indicate the line numbers where we implemented the changes that are highlighted in red in the new manuscript file.

Reviewer #2

The authors have done a good job responding to the critique. Only remaining comment is that the labeling of the scheme shown in figure 2f could be better.

We have modified the labelling by annotating a couple of missing items.

Reviewer #3

Thank you for addressing my previous concerns. With the additional evidences provided by the authors, this manuscript is much strongly presented. Couple things that need further verification: 1) line 177-178, why are different statistics used for ZT5 and ZT17 experiments? Are they both paired comparison?

We realize that the text was confusing and that the tests were not properly described. In panel b, we show the change of intracellular Cl⁻ caused by bumetanide treatment. Here, the data were paired as we measured [Cl⁻] before and after bumetanide in the same cells and the statistical significance of the Cl⁻ change was tested with the Wilcoxon signed paired test. Panel 2c show the effect of bumetanide on the entire cell population imaged at ZT17. The corrected text is now at lines 175-181.

2) line 182, Signed rank test is designed for paired test, but the ZT17 pre and post bumetanide cell number are different.

We apologize for this mistake. The test used here is the Mann-Whitney and the fixed text is now at line 180.

3) The prolonged dark exposure does answer that the reduction in intracellular Cl⁻ is not triggered by light. But since this is not a complete sleep deprivation, some amount of sleep was still preserved though they did show activity level was increased in these DE mice. Therefore, they still can't conclude that the change in Cl⁻ is mainly time-dependent.

We respectfully disagree with the reviewer about this point, since the distributions of [Cl⁻]_i in the two experimental groups are virtually identical, and yet the behavioural activity patterns are quite different.

They should at least discuss the potential interplay between circadian and sleep on such regulation.

The point at issue, as far as we can tell, is an apparent difference between our data and that reported by Alfonso et al published in Nature Neuroscience earlier in the year. They used a forced wakefulness regime, which is far more stressful than the protocol what we used and found [Cl⁻]_i maintaining the high nighttime level, whereas we saw [Cl⁻]_i drop. We believe that these data are fully compatible in view of the very different environmental manipulations that the two studies employed. We agree with the Reviewer that it is important to mention these

differences. We have now modified the Result section at lines 164-167 and the Discussion at lines 334-339.